# The transcription factor EHF promotes the maturation and immunosuppression of conventional dendritic cells

Xiaoli Liu[1,2,3], Ling Wang[3], Yongfang Xiao[3], Hai Ni[3], Jiancheng Huang [3], Kai Yao[3], Wei Zhao [1,4], Jian-Rong Yang [5], Jijun Zhao [6], Angela R. Wu [7] & Cliff Y. Yang [3] ✉

The transcriptional program that regulates immunosuppression in CCR7+ conventional dendritic cells (cDCs) is currently unknown. Here, we identify ETS homologous factor (EHF) as a transcription factor that regulates cDC maturation and immunosuppression after TLR7/8/9 stimulation. Mice with conditional deletion of EHF in DCs exhibit increased resistance to auto-immune, infection or tumor challenge. EHF-deficient DCs promotes Th1- and Th17-biased CD4+ helper T cell response in vivo and in vitro. EHF-deficient cDC1s and cDC2s exhibit decreased expression of CCR7, CD200 and PD-L1, increased expression of DC-lineage transcriptional factor IRF4, and decreased expression of inhibitory NFκB family member Rel. EHF overexpression in DCs results in the opposite phenotype. CUT&TAG analysis suggests that EHF directly regulate *Ccr7*, *Cd200*, *Cd274*, *Irf4* and *Rel* expression. Additionally, single-cell RNA-sequencing demonstrates that *Ehf* expression is highly enriched in CCR7hi DCs in mice and humans. Our study thus reveals a conserved transcriptional program that regulates cDC maturation and immunosuppression.

Dendritic cells (DCs) are indispensable for both innate and adaptive immunity. DCs could be generally categorized as type 1 (cDC1s), type 2 (cDC2As and cDC2Bs), type 3 (DC3s), plasmacytoid DCs (pDCs), transitional DCs (tDCs) and monocyte-derived DCs (moDCs)[1–9]. These subsets were differentially regulated by lineage-specific transcription factors such as BATF3, IRF4, IRF8, E2-2 and T-bet[10–21]. In addition to their importance in antigen presentation and general inflammation, DCs are also crucial for shaping immunological tolerance in many autoimmune diseases, such as cardiac and neurological autoimmunity, psoriasis, type I diabetes, and systemic lupus erythematosus (SLE)[22].

Upon maturation, CCR7+ DCs undergo a rapid transition from a pro-inflammatory state to an anti-inflammatory state. In the thymus, tolerogenic DCs are involved in the negative selection of T cells[23]. During the steady state, migratory CCR7+ DCs in the lymph nodes are found to be more immunosuppressive than resident DCs, as indicated by their superior ability to prime Foxp3+ CD4+ regulatory T cells (Tregs)[24]. Subsequently, a highly tolerogenic cDC subset (mregDCs) is identified in tumor-associated tissue[13]. mregDCs share many characteristic markers, such as CD200, FSCN1 and CCR7, with mature migratory DCs[25–27]. A similar regulatory CD103int DC subset is also

[1]Guangdong Cardiovascular Institute, Guangdong Provincial People's Hospital, Guangdong Academy of Medical Sciences, Guangzhou, China. [2]Medical Research Institute, Guangdong Provincial People's Hospital (Guangdong Academy of Medical Sciences), Southern Medical University, Guangzhou, China. [3]Department of Immunology and Microbiology, Zhongshan School of Medicine, Sun Yat-sen University, Guangzhou, Guangdong, China. [4]Key Laboratory of Stem Cells and Tissue Engineering (Sun Yat-sen University), Ministry of Education, Guangzhou, China. [5]Department of Genetics and Biomedical Informatics, Zhongshan School of Medicine, Sun Yat-sen University, Guangzhou, Guangdong, China. [6]Department of Rheumatology and Immunology, The First Affiliated Hospital, Sun Yat-sen University, Guangzhou, China. [7]Division of Life Science, Department of Chemical and Biological Engineering, The Hong Kong University of Science and Technology, Kowloon, Hong Kong, China. ✉e-mail: yangkeli6@mail.sysu.edu.cn

found in the intestinal epithelium and lungs of neonatal mice[28–30]. However, it is unclear which transcription factor specifically regulates the tolerogenic function of mregDCs.

The E26 transformation-specific (ETS) transcription factor family plays a central role in DC development. ETS family members possess a conserved winged helix-turn-helix DNA binding domain that recognizes the core DNA sequence GGA(A/T)[31]. In myeloid development, mutual repression between the ETS family member PU.1 (*Spi1*) and GATA1 regulates the divergence of megakaryocyte/erythrocyte progenitors from common myeloid progenitors[32–35]. In MoDCs, PU.1 binds to ETS motifs on its − 50 kb enhancer downstream of the *Irf8* transcriptional start site. In pDCs, the ETS family member Spi-B plays a critical role in their late development and function[36].

ETS homologous factor (EHF, also known as ESE-3) is first identified in its role on the differentiation of epithelium cells[37]. EHF is also reported to be overexpressed in human cancer cells and may promote tumor growth[38]. In the immune system, EHF is first implicated in human MoDCs in vitro[39]. Knockdown of *EHF* by RNA interference impairs surface marker expression, such as CD1a, in human moDCs generated in vitro[39]. *EHF* expression is upregulated upon stimulation of MoDCs by various ligands such as TLR ligands, cytokines, and peroxisome proliferator-activated receptor gamma agonist[40,41]. Besides DCs, *Ehf* overexpression by lentiviruses impairs the IgE-related function of murine bone-marrow-cultured mast cells[42]. Due to the lack of genetic animal tools, the immune function of EHF, especially in DCs, remains poorly defined.

Here, we report that the ETS family transcription factor EHF orchestrates an immunosuppressive program in cDC1s and cDC2s after detection of self-nucleic acids via TLR7/8/9. EHF deletion limits while its overexpression promotes cDCs immunosuppression function both in vitro and in vivo. This finding could be potential of use for treating autoimmune diseases, cancer and pathogenic infections.

## Results

### EHF maintains tolerance to colitis, and suppresses immunity to bacterial infections and cancer

To study the function of EHF in DCs in vivo, we generated transgenic mice with conditional genetic deletion of the *Ehf* allele (*Ehf* fl/fl) (Supplementary Fig. 1a). Since the *Zbtb46* gene shares the same chromosome 2 as the *Ehf* gene, it was difficult to cross *Zbtb46-Cre* with *Ehf* fl/fl mice. Thus, we chose to utilize *Itgax-Cre*, which is constitutively expressed in all DCs but also in some macrophages. In the *Ehf* fl/fl; *Itgax-Cre*+ mice (hereafter referred to as *Ehf* ΔCD11C), qPCR results demonstrated that *Ehf* was deleted in cDCs from *Ehf* ΔCD11C mice (Supplementary Fig. 1b). In general, the *Ehf* ΔCD11C mice were born at Mendelian ratios, had normal immune development, including T cells and DCs, and appeared healthy for at least 9 months (Supplementary Fig. 1c–h). Immature EHF-deficient cDC1s and cDC2s also had normal surface marker expression during steady-state (Supplementary Fig. 1i). In addition, the in vitro proinflammatory cytokine response by activated cDCs, pDCs and macrophages from *Ehf* ΔCD11C mice was normal (Supplementary Fig. 1j).

To determine the role in EHF in autoimmunity, we fed *Ehf* fl/fl and *Ehf* ΔCD11C mice dextran sodium sulfate (DSS)-containing water, which can damage the epithelial monolayer of the large intestine, as a model for colitis. After a typically nonlethal 2.5% dose of DSS, none of the *Ehf* ΔCD11C mice survived beyond 16 days, while 85% of the *Ehf* fl/fl mice survived (Fig. 1a). Correspondingly, the body weight of the *Ehf* ΔCD11C mice decreased more rapidly than that of the *Ehf* fl/fl mice in the first week (Fig. 1b). The disease activity index (DAI), which measures clinical colitis symptoms, was substantially greater in *Ehf* ΔCD11C mice than in *Ehf* fl/fl mice (Fig. 1c). Analysis of the inflamed colons revealed that, compared with *Ehf* fl/fl mice, *Ehf* ΔCD11C mice had shorter colon lengths, more infiltrated immune cells and higher histological H&E scores in colon sections (Fig. 1d–f). While DSS induces colitis in a T-cell independent

manner, studies had shown that T-cell responses contribute to the inflammatory response after colitis initiation[43–47]. Correspondingly, we found that overall CD4+ and CD8+ T cells exhibited higher levels of activation marker CD44 in the DSS-treated *Ehf* ΔCD11C mice (Supplementary Fig. 1k). PD-1 expression of CD4+ T cells in the DSS-treated *Ehf* ΔCD11C mice were also lower (Supplementary Fig. 1l). The IFNγ- and IL-17A-producing capacities of activated CD4+ T cells were also increased in DSS-treated *Ehf* ΔCD11C mice (Supplementary Fig. 1m). Here, we demonstrated that EHF suppressed autoimmunity in a DSS-induced colitis model.

Next, we examined the role of EHF in bacterial, viral infections and cancer in vivo (Fig. 1g). First, we intravenously injected mice with a lethal dose of the intracellular bacteria *Listeria monocytogenes*. *Ehf* fl/fl mice reached ~55% mortality on Day 6 post-infection, while all the *Ehf* ΔCD11C mice survived (Fig. 1h). Correspondingly, we found that all CD4+ T cells, but not CD8+ T cells, exhibited higher levels of activation marker CD44 in the infected *Ehf* ΔCD11C mice (Supplementary Fig. 1n). The IL-2, IFNγ and TNF producing capacities of CD44+CD4+ T cells were also increased in the *Ehf* ΔCD11C mice after *L. monocytogenes* infection (Supplementary Fig. 1o). Second, we infected mice with vesicular stomach virus (VSV). There was no difference in viral titers between *Ehf* fl/fl and *Ehf* ΔCD11C mice (Fig. 1i). However, CD4+ and CD8+ T cells exhibited higher levels of activation marker CD44 in the infected *Ehf* ΔCD11C mice (Supplementary Fig. 1p). The IL-2, IFNγ and TNF producing capacities of CD44+CD4+ T cells were also increased in the *Ehf* ΔCD11C mice after VSV infection (Supplementary Fig. 1q). Third, after transplantation with highly lethal B16 melanoma cells with recombinant ovalbumin (B16-OVA), *Ehf* ΔCD11C mice had a greater overall survival rate than *Ehf* fl/fl mice (Fig. 1j). The CTL infiltration in the tumor increased ~ 2-fold (Fig. 1k). Following injection of OVA-specific TCR-transgenic CD8+ OT-I cells 11 days after B16 transplantation, the OT-I cells in the draining lymph nodes from the *Ehf* ΔCD11C mice expressed lower levels of PD-1 after 3 days (Fig. 1l). Here we showed that EHF repressed immune response to bacterial infections and melanoma xenograft in mice.

### EHF suppresses Th1- and Th17-biased CD4+ T-cell response

Since CD4+ and CD8+ T cells in our disease models all exhibited a more activated phenotype in the EHF-deficient mice, we specifically examined the in vivo CD4+ or CD8+ T-cell priming ability of EHF-deficient DCs (Fig. 2a and Supplementary Fig. S2a). Using CpG-B as a Th1-biased adjuvant, we found that the percentage and number of OVA-specific OT-II cells in the spleen and draining lymph nodes were increased by approximately 2-fold in *Ehf* ΔCD11C mice (Fig. 2b). Correspondingly, we found that the IL-2, TNF, IFNγ and T-bet expression by OT-II cells were increased in *Ehf* ΔCD11C mice (Fig. 2c). To determine the role of EHF in the Th2-biased response, we utilized papain and IFA as adjuvants, and found that the overall OT-II ratio, number, IL-4 production and GATA-3 expression remained unchanged between *Ehf* fl/fl and *Ehf* ΔCD11C mice (Fig. 2d and Supplementary Fig. S2b). To determine the role of EHF in follicular helper T cells (Tfh), we utilized sheep red blood cells associated with OVA (SRBC-OVA). We found that EHF was dispensable for the expansion of OT-II cells, Tfh and germinal center B cells (Fig. 2e and Supplementary Fig. S2c). Here, we demonstrate that EHF specifically suppresses Th1 response in vivo.

To determine the role of EHF in cross-presentation, we adoptively transferred OT-I cells before injecting mice with cell-associated OVA[48–50]. We found that EHF is dispensable for CTL cross priming (Fig. 2f and Supplementary Fig. S2d). To determine the role of EHF in the overall CD8+ T-cell priming, we adoptively transferred OT-I cells before challenging mice with a sublethal dose of *Listeria monocytogenes* with recombinant ovalbumin (LM-OVA). We found that EHF is dispensable for CTL expansion and cytokine production (Supplementary Fig. 2e, f). Our results showed that EHF in DCs is dispensable for the antigen-specific CD8+ T-cell response in vivo.

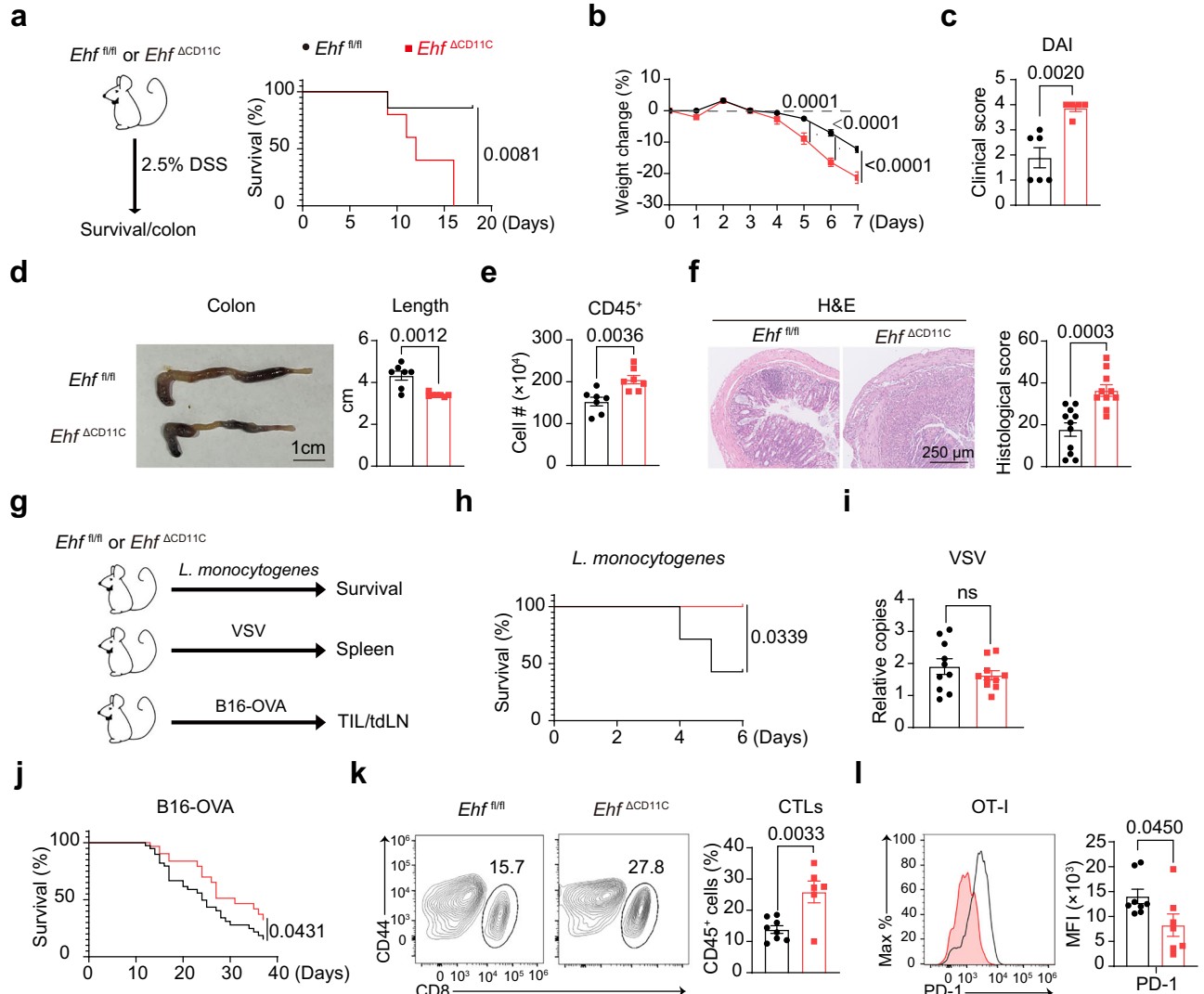

**Fig. 1 | EHF maintains tolerance to colitis, and suppresses immunity to bacterial infections and cancer. a** The survival rate of *Ehf*[fl/fl] or *Ehf*[ΔCD11C] mice after 2.5% DSS administration is shown (*Ehf*[fl/fl], *n* = 7 mice; *Ehf*[ΔCD11C], *n* = 5 mice). **b** Changes in the weights of the *Ehf*[fl/fl] and *Ehf*[ΔCD11C] mice after continuous administration of 2.5% DSS in drinking water are shown (*n* = 6 mice per group). **c** The disease activity index (DAI) of *Ehf*[fl/fl] or *Ehf*[ΔCD11C] mice were measured on Day 7 after 2.5% DSS treatment (*Ehf*[fl/fl], *n* = 6 mice; *Ehf*[ΔCD11C], *n* = 5 mice). **d** Representative images of the colons (left) from *Ehf*[fl/fl] or *Ehf*[ΔCD11C] mice, their lengths (right) after 7 days of DSS treatment are shown (*n* = 7 mice per group; scale bar = 1 cm). **e** Infiltrated CD45[+] immune cells in the colon after 7 days of 2.5% DSS treatment are shown (*n* = 7 mice per group). **f** Representative images (left) and histological colitis score (right) of H&E colon sections from *Ehf*[fl/fl] or *Ehf*[ΔCD11C] mice after 7 days of 2.5% DSS treatment is shown (*Ehf*[fl/fl], *n* = 11 mice; *Ehf*[ΔCD11C], *n* = 10 mice; scale bar = 250 μm). **g** *Ehf*[fl/fl] or *Ehf*[ΔCD11C] mice were subjected to *L. monocytogenes,* vesicular stomach virus (VSV) infections

or B16-OVA xenograft. **h** The survival rate of *Ehf*[fl/fl] or *Ehf*[ΔCD11C] mice after lethal challenge with LM is shown (*Ehf*[fl/fl], *n* = 7 mice; *Ehf*[ΔCD11C], *n* = 6 mice). **i** The splenic viral load in VSV-infected *Ehf*[fl/fl] or *Ehf*[ΔCD11C] mice was measured via qPCR on Day 5 (*n* = 10 mice per group). **j** The survival rate of *Ehf*[fl/fl] or *Ehf*[ΔCD11C] mice after xenografting with B16-OVA melanoma cells is shown (*Ehf*[fl/fl], *n* = 39 mice; *Ehf*[ΔCD11C], *n* = 31 mice). **k** Total CTL infiltration in tumors was measured via FACS (*Ehf*[fl/fl], *n* = 8 mice; *Ehf*[ΔCD11C], *n* = 6 mice). **l** On Day 11 after B16-OVA transplantation, OT-I cells were i.v. injected. PD-1 expression in OT-I cells in the tumor-draining lymph nodes was measured via FACS after 3 days (*Ehf*[fl/fl], *n* = 8 mice; *Ehf*[ΔCD11C], *n* = 7 mice). *P*-values were calculated by two-tailed Student's *t* test except for (**a**, **h** and **j**) (log-rank Mantel-Cox test) and **b** (two-way ANOVA with Sidak's multiple comparisons test) (ns, *P* > 0.05; *, *P* < 0.05; **, *P* < 0.01; ***, *P* < 0.001; ****, *P* < 0.0001). All the data are presented as the means ± SEMs and are representative of three independent experiments.

In the DSS colitis model, we found that IL-17A production by CD44[+]CD4[+] T cells were increased in EHF-deficient mice (Supplementary Fig. 1m). To further examine Th17 response in vivo, we infected mice with *Citrobacter rodentium*, an extracellular bacteria that infects the large intestine and causes colon inflammation[51]. Compared with *Ehf*[fl/fl] mice, *Ehf*[ΔCD11C] mice had longer colon lengths and decreased immune cell numbers in the mesenteric lymph nodes, suggesting a less severe colon inflammation in *Ehf*[ΔCD11C] mice (Supplementary Fig. 2g). IL-17A and IL-22 production by CD44[+]CD4[+] T cells from mesenteric lymph nodes and colons also increased in *Ehf*[ΔCD11C] mice, suggesting a stronger type III immune response (Supplementary Fig. 2h). The IL-2, IFNγ, and TNF-producing

capacities of CD44[+]CD4[+] T cells were increased in the *Ehf*[ΔCD11C] mice (Supplementary Fig. 2i). Here we found that Th17 response were enhanced in the *Ehf*[ΔCD11C] mice in vivo.

Lastly, we FACS-sorted cDC1s or cDC2s, pulsed them with TCR-specific OVA peptides, and incubated them with FACS-sorted OT-II or OT-I cells with different polarization cytokines in vitro. We found that EHF-deficient cDC1s and cDC2s primed increased IFNγ-producing Th1 and IL-17A-producing Th17 cells (Fig. 2g). In comparison, EHF in cDC1s or cDC2s was dispensable for IL-4-producing Th2 cells or inducible regulatory T cells (Fig. 2g). Natural regulatory T cells were also normal in EHF-deficient mice (Supplementary Fig. 1h). Consistently, EHF in cDC1s or cDC2s was dispensable for CTL priming in vitro (Fig. 2g).

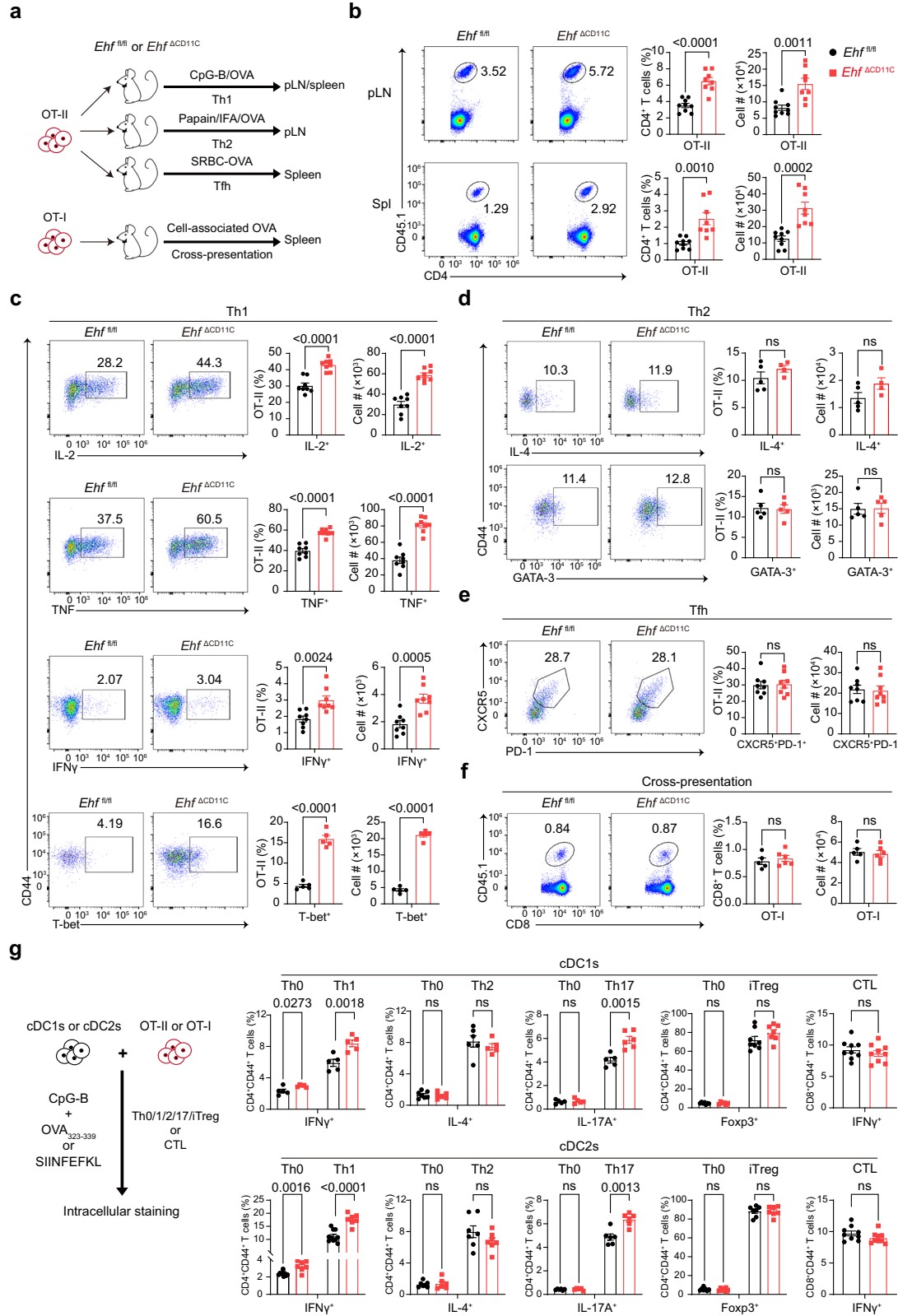

Here, we found that EHF in cDC1s or cDC2s suppresses Th1 and Th17 priming in vitro.

## EHF is upregulated by TLR7/8/9 and suppressed by TLR3/GM-CSF/IFNγ

Under steady state in C57BL/6 J (B6 or WT) mice, *Ehf* expression was moderately enriched in cDCs, but undetectable in FACS-sorted pDCs, B cells, CD4[+]/CD8[+] T cells, monocytes, neutrophils and macrophages (Supplementary Fig. 3a). We FACS-sorted cDCs and stimulated them in vitro with various ligands. We found that cDC1s and cDC2s upregulated *Ehf* expression by ~10-fold after TLR9 (CpG-B) stimulation, and by ~5-fold after TLR7/8 (R848) stimulation (Fig. 3a). In contrast, the TLR3 agonist poly(I:C) downregulated *Ehf* expression by ~3-fold (Fig. 3a).

**Fig. 2 | EHF suppresses Th1- and Th17-biased CD4$^+$ T-cell response.**
**a** Experimental design of Th1, Th2, and Tfh CD4$^+$ T cell priming and antigen cross-presentation assay with cell-associated OVA in vivo. **b** CD45.1$^+$ OT-II expansion in popliteal lymph nodes and spleens from $Ehf^{fl/fl}$ and $Ehf^{\Delta CD11C}$ mice was measured by FACS on Day 5 after s.c. injection of CpG-B and OVA ($Ehf^{fl/fl}$, $n = 9$ mice; $Ehf^{\Delta CD11C}$, $n = 8$ mice). **c** IL-2, TNF, IFNγ and expression by restimulated OT-II cells were measured by FACS on Day 5 after s.c. injection of CpG-B and OVA (IL-2, TNF and IFNγ, $n = 8$ mice per group; T-bet, $n = 5$ mice per group). **d** IL-4 and GATA-3 expression by restimulated OT-II cells from popliteal lymph nodes was measured by FACS on Day 5 after s.c. injection of papain/IFA/OVA (IL-4: $Ehf^{fl/fl}$, $n = 5$ mice; $Ehf^{\Delta CD11C}$, $n = 4$ mice; GATA-3: $n = 5$ mice per group). **e** Splenic CXCR5$^+$PD-1$^+$ OT-II cells were measured by FACS 5 days after i.p. injection of SRBC-OVA ($n = 8$ mice per group). **f** CD45.1$^+$ OT-I cell expansion in spleens from $Ehf^{fl/fl}$ and $Ehf^{\Delta CD11C}$ mice was measured by FACS on Day 5 after infection with cell-associated OVA ($Ehf^{fl/fl}$, $n = 5$ mice; $Ehf^{\Delta CD11C}$, $n = 6$ mice). **g** Splenic cDC1s and cDC2s from $Ehf^{fl/fl}$ and $Ehf^{\Delta CD11C}$ mice were FACS-sorted, pulsed with CpG-B and OVA$_{323-339}$ or SIINFEKL, then incubated with FACS-sorted CD45.1$^+$ OT-II or OT-I cell under Th0 (PBS), Th1 (IL-12p70), Th2 (IL-4), Th17 (TGF-β, IL-6 and IL-1β), iTreg (TGF-β1 and IL-2) or CTL (PBS) conditions for 5 days. IFNγ, IL-4, IL-17A and Foxp3 expression were measured by intracellular staining (cDC1s: $n = 5$ mice per group (IFNγ$^+$ under Th0 and Th1); $Ehf^{fl/fl}$, $n = 7$; $Ehf^{\Delta CD11C}$, $n = 6$ (IL-4$^+$ under Th0); $Ehf^{fl/fl}$, $n = 6$, $Ehf^{\Delta CD11C}$, $n = 5$ mice (IL-4$^+$ under Th2); $n = 5$ mice per group (IL-17A$^+$ under Th0); $Ehf^{fl/fl}$, $n = 5$; $Ehf^{\Delta CD11C}$, $n = 6$ (IL-17A$^+$ under Th17); $Ehf^{fl/fl}$, $n = 7$; $Ehf^{\Delta CD11C}$, $n = 6$ (Foxp3$^+$ under Th0); $Ehf^{fl/fl}$, $n = 8$; $Ehf^{\Delta CD11C}$, $n = 7$ (Foxp3$^+$ under iTreg); $Ehf^{fl/fl}$, $n = 10$; $Ehf^{\Delta CD11C}$, $n = 8$ (IFNγ$^+$ under CTL). cDC2s: $Ehf^{fl/fl}$, $n = 10$; $Ehf^{\Delta CD11C}$, $n = 7$ (IFNγ$^+$ under Th0 and Th2); $Ehf^{fl/fl}$, $n = 7$; $Ehf^{\Delta CD11C}$, $n = 6$ (IL-4$^+$ under Th0); $n = 7$ mice per group (IL-4$^+$ under Th2); $Ehf^{fl/fl}$, $n = 6$, $Ehf^{\Delta CD11C}$, $n = 5$ (IL-17A$^+$ under Th0); $n = 6$ mice per group (IL-17A$^+$ under Th17); $n = 7$ mice per group (Foxp3$^+$ under Th0 and iTreg); $Ehf^{fl/fl}$, $n = 10$; $Ehf^{\Delta CD11C}$, $n = 8$ mice (IFNγ$^+$ under CTL)). $P$- values were calculated by two-tailed Student's $t$ test (ns, $P > 0.05$; *, $P < 0.05$; **, $P < 0.01$; ***, $P < 0.001$; ****, $P < 0.0001$). All the data are presented as the means ± SEMs and are representative of three independent experiments.

Neither TLR2 nor TLR4 agonizts altered $Ehf$ expression in cDCs (Fig. 3a).

To determine if any immune cells besides cDCs express EHF after TLR9 stimulation, we FACS-sorted cDC1s/cDC2s, pDCs/tDCs, monocytes, macrophages, neutrophils, B and T cells from PBS or CpG-B injected B6 mice, and then performed single-cell RNA-sequencing (scRNA-seq) with the pooled populations (Fig. 3b and Supplementary Fig. S3b). Consistently, we found that $Ehf$ expression was only present in CCR7$^{hi}$ DCs (Fig. 3b). In CpG-B injected WT mice, we FACS-sorted cDC1s, cDC2As, cDC2Bs, DC3s, and moDCs, and examined $Ehf$ expression after CpG-B stimulation in vivo[10,12,52]. All cDC1s and cDC2 subsets upregulated $Ehf$ expression (Fig. 3c). However, DC3s and moDCs did not appear to express $Ehf$ (Fig. 3c).

Since "regulatory DCs" are reported to be triggered by the uptake of apoptotic cells, we incubated FACS-sorted WT cDCs with apoptotic WT splenic cells irradiated with a lethal dose of UV light[28,29]. We found that $Ehf$ expression was upregulated ~ 4-fold by apoptotic cells (Fig. 3d). The EHF upregulation by CpG-B or apoptotic cells was blocked if antigen uptake was inhibited, suggested that it is dependent on antigen uptake (Fig. 3e)[53]. In MyD88-deficient mice, $Ehf$ upregulation by TLR7/8/9 or apoptotic cells in cDCs was completely abolished, suggesting that the upregulation of EHF was mediated directly via the canonical TLR signaling pathway (Fig. 3f). After CpG-B treatment, there were significant differences in CCR7, CD200 and PD-L1 expression in DCs from MyD88-deficient mice (Supplementary Fig. 3c, d).

The in vivo upregulation of $Ehf$ in cDCs peaked approximately 12-48 hrs after CpG-B injection before $Ehf$ was downregulated (Fig. 3g). To identify factors that may downregulate EHF expression, we screened 23 different cytokines in CpG-B-activated cDCs. We found that only the addition of GM-CSF and IFNγ repressed $Ehf$ expression in cDCs (Fig. 3h). When GM-CSF and IFNγ neutralizing antibodies were used in vivo, $Ehf$ expression in CpG-B-activated cDCs showed further upregulation (Fig. 3i). Consistently, $Ehf$ expression in GM-CSF-cultured BMDCs was not detected by qPCR, even after CpG-B activation (Supplementary Fig. 3e). In comparison, cDC$^{FL-Notch}$ cells, which were cultured from bone marrow cells with FLT3L and OP9 stromal cells expressing the Notch ligand Delta-like 1 (OP9-DL1), had a robust $Ehf$ expression before or after CpG-B expression (Supplementary Fig. 3e)[54,55]. Here, we showed that GM-CSF and IFNγ could block EHF upregulation in cDCs.

### EHF promotes CCR7, CD200 and PD-L1 expression in cDC1s and cDC2s

Since EHF is upregulated the most by TLR9 activation, we investigated the role of EHF in DC maturation by injecting CpG-B into $Ehf^{fl/fl}$ or $Ehf^{\Delta CD11C}$ mice. We found that ~ 25% less MHCII$^+$CD11c$^+$ DCs in the spleen of $Ehf^{\Delta CD11C}$ mice at 12 h post-injection (Fig. 4a). This deficiency was primarily from the compartment of CCR7$^+$CD200$^+$ DCs (Fig. 4b). Since DC development and cell numbers were normal in EHF-deficient mice, this deficiency in cell numbers could be due to a defect in trafficking (Supplementary Fig. 1c, d). In a transwell-based migration assay, we found that EHF-deficient cDC1s and cDC2s had a defect in trafficking to CCR7 ligands CCL19/CCL21 in a dose-dependent manner (Supplementary Fig. 4a).

Besides traditional cDC1s and cDC2s, we performed a more detailed analysis of DC subsets. The cell number defect is consistent in cDC1s, cDC2As and cDC2Bs (Fig. 4c). CCR7, CD200 and PD-L1 expression in cDC1s, cDC2As, cDC2Bs and DC3s were significantly downregulated in $Ehf^{\Delta CD11C}$ mice after CpG-B stimulation (Fig. 4d). This PD-L1 downregulation is consistent with the lower expression of PD-1 on OT-I cells in the tumor model and CD4$^+$ T cells in the DSS model (Fig. 1l and Supplementary Fig. S11). Other surface markers in TLR9-stimulated EHF-deficient cDCs, macrophages and pDCs were normal (Supplementary Fig. 4b and S4c). In the mesenteric lymph nodes of DSS-treated mice, we also found that EHF-deficient migratory cDC1s and cDC2s downregulated CCR7, CD200 and PD-L1 expression (Fig. 4e and Supplementary Fig. S4d).

Next, we measured DC-lineage transcriptional factor IRF4 and IRF8 expression in CpG-B-stimulated $Ehf^{fl/fl}$ and $Ehf^{\Delta CD11C}$ cDCs via intracellular staining[5]. In $Ehf^{\Delta CD11C}$ mice, IRF4 expression was upregulated in cDC1s and cDC2s (Fig. 4f). In comparison, IRF8 expression remained unchanged in cDC1s or cDC2s regardless of EHF-deletion (Supplementary Fig. 4e). The NFκB transcriptional family members Rel and RelB are known to dampen the proinflammatory NFκB response in DCs[56–58]. We found that only Rel (detected by its subunit c-Rel) expression were downregulated in CpG-B-activated $Ehf^{\Delta CD11C}$ cDC1s and cDC2s (Fig. 4f and Supplementary Fig. S4e). In addition, over-expressing EHF in cDC$^{FL-Notch}$ via retroviruses upregulated CCR7, CD200 and PD-L1 expression, downregulated IRF4 expression, while upregulating Rel level (Fig. 4g). These experiments demonstrated that EHF regulated the expression of CCR7, CD200 and PD-L1 in cDC1s and cDC2s.

### CUT&TAG analysis suggests that EHF may directly regulate gene transcription in DCs

To determine the genetic targets of EHF via chromatin-based sequencing analysis, we generated anti-EHF monoclonal antibodies by vaccinating mice with a partial peptide of EHF that contains its DNA-binding domain (Supplementary Fig. 5a). After several rounds of vaccination, we selected one clone (clone number: XL-32) from multiple positive hybridomas. Based on Western blotting, this clone was able to distinguish EHF expression in different organs and EHF overexpression in 293 T cells (Supplementary Fig. 5b, c). Most importantly, this clone detected significantly fewer anti-EHF signals from CpG-B-treated $Ehf^{\Delta CD11C}$ DCs, thus demonstrating the specificity of this clone for EHF (Supplementary Fig. 5d).

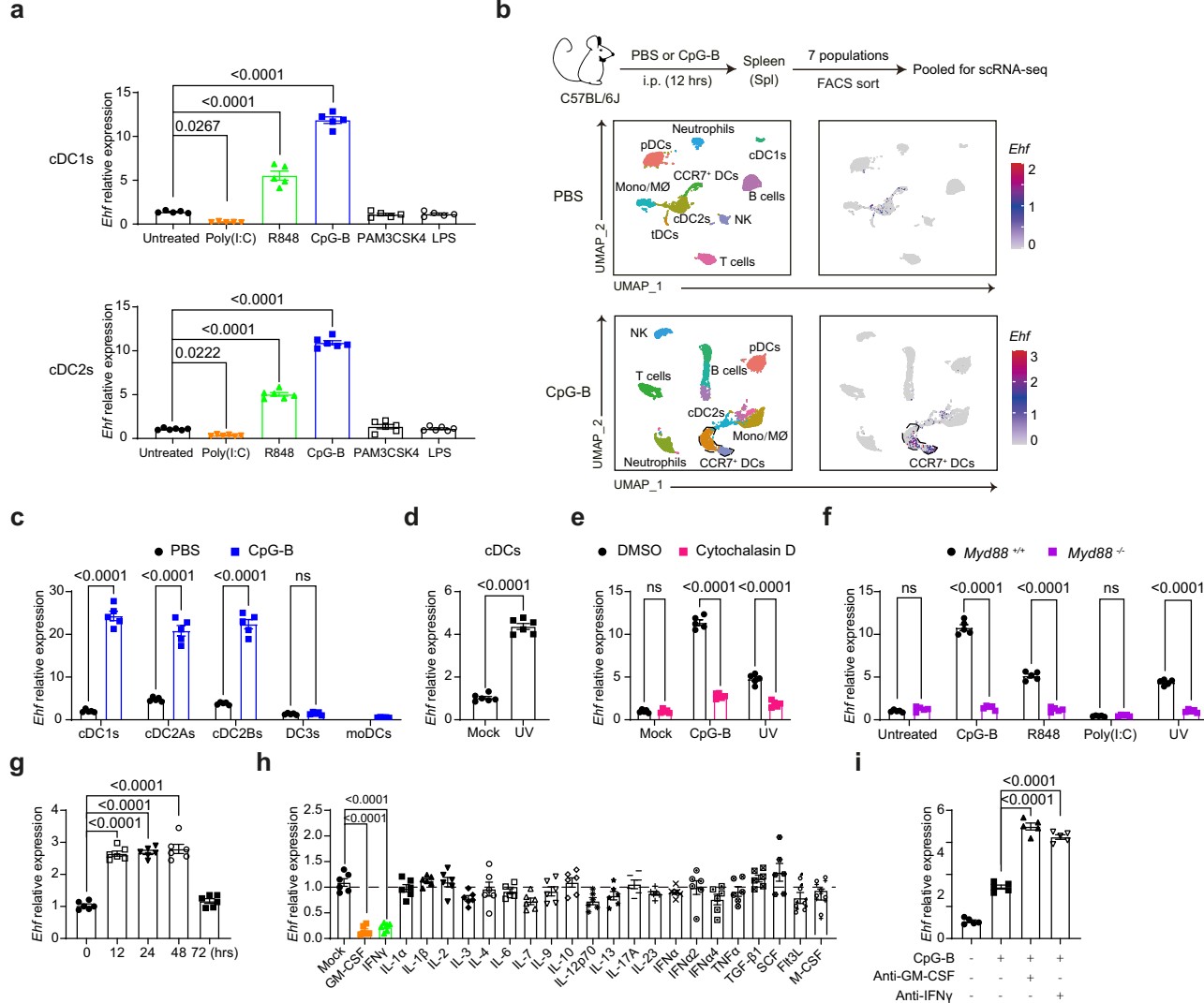

**Fig. 3 | EHF is upregulated by TLR7/8/9 and suppressed by TLR3/GM-CSF/IFNγ.**
**a** cDC1s and cDC2s were FACS-sorted from B6 mouse spleens, and *Ehf* transcript levels were measured via qPCR after in vitro stimulation with indicated ligands for 14 hrs (cDC1s, *n* = 5; cDC2s, *n* = 6 independent runs with 10 pooled mice). **b** After PBS or CpG-B injection, cDCs, pDCs, macrophages, monocytes, neutrophils, B and T cells were FACS-sorted and pooled for scRNA sequencing. UMAP dimensionality reduction on the individual dataset (left) and feature plots depicting *Ehf* expression (right) are shown. **c** cDC1s, cDC2As, cDC2Bs, cDC3s and moDCs were FACS-sorted from B6 spleens after PBS or CpG-B injection and *Ehf* transcript levels were measured via qPCR (*n* = 5 independent runs with 10 pooled mice). **d** FACS-sorted cDCs from B6 mouse spleens were stimulated in vitro with UV-irradiated B6 splenic cells for 14 h, and *Ehf* transcript levels were measured via qPCR (*n* = 6 independent runs with 10 pooled mice). **e** FACS-sorted cDCs from B6 spleens were treated with cytochalasin D for 30 min in vitro, and then stimulated with CpG-B and UV-irradiated B6 splenic cells for 14 h. *Ehf* transcript levels were measured via qPCR (*n* = 5 independent runs with 10 pooled mice). **f** *Myd88*[+/+] or *Myd88*[-/-] splenic cDCs

were FACS-sorted, and *Ehf* transcript levels were measured via qPCR after in vitro stimulation with the indicated TLR ligands for 14 hrs (*n* = 5 independent runs with 10 pooled mice per group). **g** The time-course of *Ehf* expression in cDCs from B6 popliteal lymph nodes after CpG-B footpad injection was measured via qPCR (*n* = 6 independent runs with 10 pooled B6 mice). **h** cDCs were FACS-sorted from B6 mouse spleens, and stimulated in vitro with the indicated cytokines for 14h. *Ehf* transcript levels were then measured via qPCR (*n* = 6 independent runs with 10 pooled mice). **i** cDCs were FACS-sorted from B6 mouse spleens, 14 h after PBS or CpG-B injection, along with or without anti-GM-CSF or anti-IFNγ monoclonal antibodies. *Ehf* transcripts were then measured via qPCR (*n* = 5 independent runs with 3 pooled mice). *P*-values were calculated by two-tailed Student's *t* test except for (**a**, **g**–**i**) (one-way ANOVA with Dunnett's multiple comparisons test) (ns, *P* > 0.05; *, *P* < 0.05; **, *P* < 0.01; ***, *P* < 0.001; ****, *P* < 0.0001). All the data are presented as the means ± SEMs and are representative of three independent experiments, except for (**b**), which was from one experiment. All qPCR data were first normalized to β-actin expression.

To determine whether EHF directly or indirectly regulates inhibitory and maturation markers in cDCs, we performed CUT&TAG analysis on CpG-B-stimulated FACS-sorted cDC[FL-Notch] cells. Isotype control IgG on *Ehf*[fl/fl] cDC[FL-Notch] and anti-EHF antibodies on *Ehf*[ΔCD11C] cDC[FL-Notch] were utilized as two different controls. Genome-wide comparison of CUT&TAG signals at -3 kb to 3 kb from transcription start site (TSS) indicated that the samples with anti-EHF antibodies exhibited a strong peak compared to the lack of peaks in the IgG sample (Fig. 5a). A weaker signal was still detected in *Ehf*[ΔCD11C] cDCs, as *Itgax-Cre* deletion could not reach 100% in DCs (Fig. 5a)[59]. Motif enrichment analysis by HOMER

showed that DNA fragments pulled down by our in-house anti-EHF monoclonal antibody clone XL-32 were exclusively enriched in ETS domains (Fig. 5b). HOMER analysis also revealed that the de novo motif for EHF contained the conserved ETS family motif GGAA (Fig. 5c). Gene ontology (GO) enrichment analysis of enriched peaks demonstrated that EHF regulated the transcription of multiple pathways such as leukocyte migration, mononuclear differentiation, bacterial response, immune activation, immune inhibition, inflammatory response, T-cell priming, PRR signaling, transcriptional factor activity, and antigen presentation (Fig. 5d and Supplementary Fig. S5e).

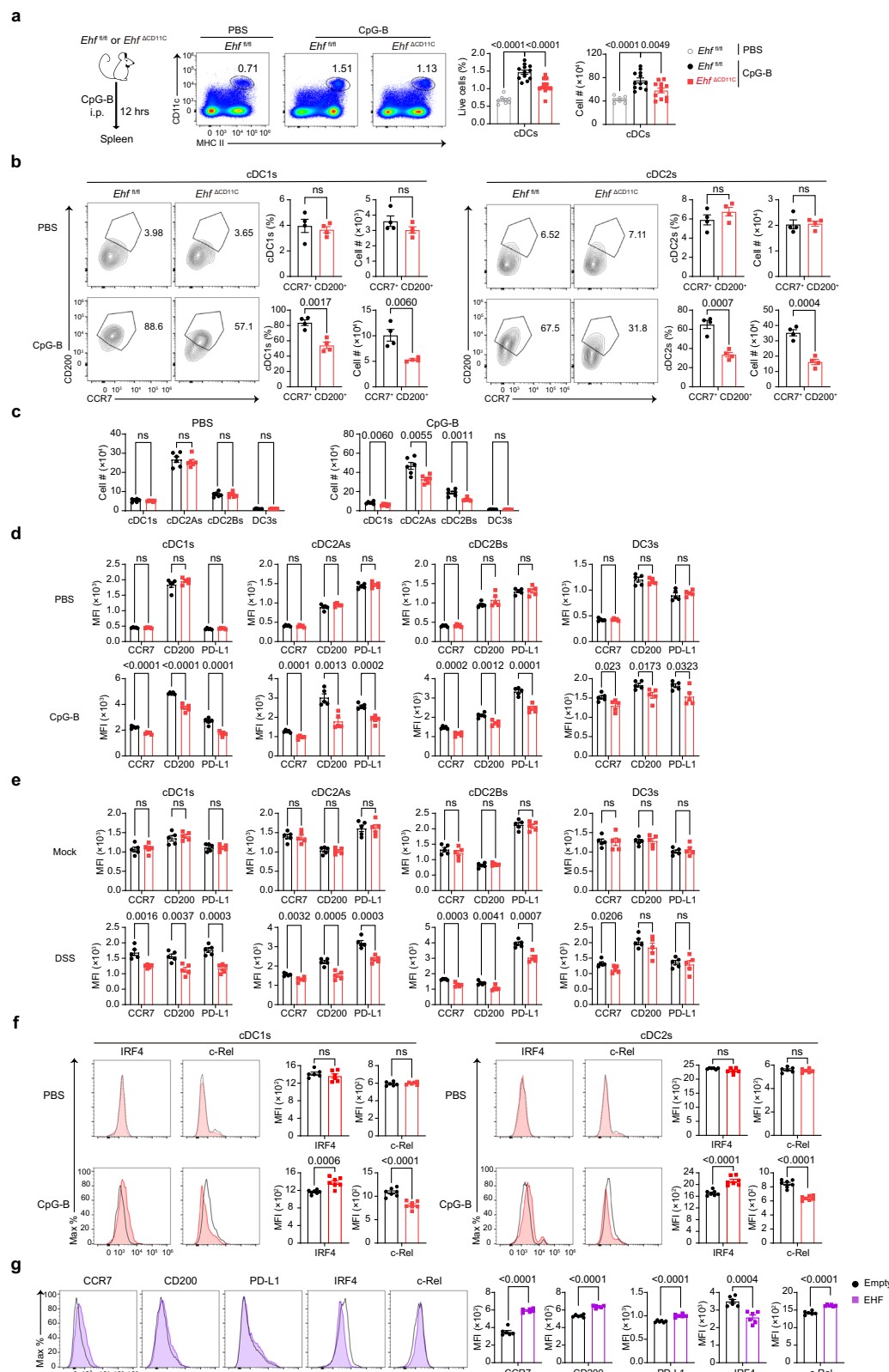

We previously demonstrated that CCR7, CD200 and PD-L1 (encoded by *Cd274*), were downregulated in EHF-deficient cDC1s and cDC2s (Fig. 4d, e). DNA-binding sites generated from CUT&TAG showed that there were multiple strong peaks in the vicinity of the *Ccr7*, *Cd200* and *Cd274* loci, while the *Ehf*^ΔCD11C sample had correspondingly reduced signals (Fig. 5e). Besides cDC^FL-Notch, we performed CUT&TAG on cDC1s and cDC2s sorted from spleens of CpG-

B-treated WT mice, and found similar peaks on the same *Ccr7*, *Cd200* and *Cd274* loci (Supplementary Fig. 5f). CUT&TAG analysis also suggested that EHF could bind to ETS domains in a 3' proximal enhancer located at +20 kb from the *Irf4* TSS (Fig. 5e and Supplementary Fig. S5f). Similarly, we detected strong peaks in the 5' promoters (~1 kb) region of *Rel* (Fig. 5e and Supplementary Fig. S5f). Within the denoted boxed peaks, we designed qPCR primers around

**Fig. 4 | EHF promotes CCR7, CD200 and PD-L1 expression in cDC1s and cDC2s.**
**a** The proportions of splenic MHCII⁺CD11c⁺cDCs in *Ehf* fl/fl and *Ehf* ΔCD11C mice and their cell numbers were measured via FACS at 12 hrs after i.p. injection of PBS or CpG-B (PBS: *Ehf* fl/fl, n = 8 mice; CpG-B, n = 12 mice per group). **b** The proportions of splenic CCR7⁺CD200⁺ cDC1s and CCR7⁺CD200⁺ cDC2s in *Ehf* fl/fl and *Ehf* ΔCD11C mice, and their cell numbers were measured via FACS at 12 h after i.p. injection of PBS or CpG-B (n = 4 mice per group). **c** The cell numbers of splenic cDC1s, cDC2As, cDC2Bs and DC3s from *Ehf* fl/fl and *Ehf* ΔCD11C mice were measured via FACS at 12 h after i.p. injection of PBS or CpG-B (n = 6 mice per group). **d** Expression of the indicated surface molecules on splenic cDC1s, cDC2As, cDC2Bs and DC3s from *Ehf* fl/fl and *Ehf* ΔCD11C mice was measured by FACS at 12 h after i.p. injection of PBS or CpG-B (n = 5 mice per group). **e** Expression of the indicated surface molecules on mesenteric

lymph nodes migratory cDC1s, cDC2As, cDC2Bs and DC3s from *Ehf* fl/fl and *Ehf* ΔCD11C mice was measured by FACS on Day 7 after mock or DSS treatment (n = 5 mice per group). **f** IRF4 or c-Rel expression in splenic cDC1s and cDC2s from *Ehf* fl/fl and *Ehf* ΔCD11C mice was measured via intracellular staining at 12 hrs after PBS or CpG-B injection (PBS, n = 6 mice per group; CpG-B, n = 7 mice per group). **g** EHF was overexpressed via retroviruses in cDC FL-Notch, and the expression of the indicated surface makers in GFP⁺ DCs was measured by FACS (left) and MFI (right) (n = 6 mice per group). P values were calculated by two-tailed Student's t test except for (**a**) (one-way ANOVA with Dunnett's multiple comparisons test) (ns, P > 0.05; *, P < 0.05; **, P < 0.01; ***, P < 0.001; ****, P < 0.0001). All the data are presented as the means ± SEMs and are representative of three independent experiments.

randomly chosen ETS motifs, and found that the enrichment measured by qPCR were consistent with the CUT&TAG sequencing data (Fig. 5f). After identifying ETS binding motifs within the enriched peaks, we performed motif mutagenesis on these motifs in luciferase reporter assays. We found that these regions with ETS motifs were critical for EHF's transcriptional activity (Fig. 5g). The combined data from genetic knockout mice, CUT&TAG analysis and in vitro studies demonstrated that EHF may directly regulate *Ccr7, Cd200, Cd274, Irf4,* and *Rel* transcription at the chromatin level.

### EHF is expressed by human and murine CCR7hi DCs
To get more clarity on which DC subset expresses EHF, we injected WT mice with the TLR9 agonist CpG-B, and then FACS-sorted MHCII⁺CD11c⁺ DCs from the spleen, lymph nodes and peritoneal cavity for single-cell RNA-sequencing (Fig. 6a and Supplementary Fig. S6a). When all six samples were subjected to integrated transcriptional analysis, CCR7⁺ DCs were separated into 3 clusters (Fig. 6b). Cluster 1 expressed intermediate levels of CCR7, while Clusters 2 and 3 expressed higher levels of CCR7 (Supplementary Fig. 6b). Among the CCR7hi DCs, the marker genes from the spleen, lymph nodes and peritoneal cavity all included *Ehf* (Supplementary Fig. 6c). In addition to exhibiting high *Ehf* expression, CCR7hi DCs (Clusters 2 and 3) also exhibited increased expression of inhibitory molecules and maturation markers such as *Cd200, Socs2, Il2ra, Cd274* (encodes PD-L1) and *Cd83*, as well as the anti-inflammatory NFκB family transcription factors *Rel* and *RelB* (Fig. 6c). *Ehf* transcripts were enriched only in CCR7hi DCs (Clusters 2 or 3) from the spleen, lymph nodes and peritoneal cavity after CpG-B injection (Fig. 6d). In summary, we found that *Ehf* expression was specifically enriched in CCR7hi DCs across different tissues after CpG-B stimulation in vivo.

To validate the EHF expression in humans, we collected PBMCs from healthy volunteers, treated the cells with PBS or CpG-B in vitro for 16 hrs, FACS-sorted CD11C⁺HLA-DR⁺ DCs, and pooled the individual samples for single-cell RNA-seq (Supplementary Fig. 6e). A previously published single-cell RNA-seq database of fresh human DCs from PBMCs was used for the untreated control data[18]. In the fresh untreated DCs, *EHF* expression was not specifically enriched in any DC subset (Fig. 6e). In the CpG-B-overnight-treated DCs, *EHF* expression was highly enriched in CCR7hi DCs (Clusters 2 and 3) but not in CCR7int DCs (Cluster 1) or other DCs (Fig. 6e and Supplementary Fig. S6f, g). In the mock-treated DCs incubated overnight, EHF was upregulated in CCR7hi DCs, perhaps due to TLR7/8/9 autoactivation by apoptotic cells (Fig. 6e). Compared to CCR7int DCs, CCR7hi DCs (Clusters 2 and 3) showed greater enrichment in inhibitory, migratory and maturation-related molecules such as *CD25, CD200, SOCS2, CD274, CCR7, FSCN1, CXCL16, CD83, REL* and *RELB* (Fig. 6f). Similar to murine data, human EHF expression was upregulated by TLR7/8/9 stimulation, but downregulated by TLR3 activation (Fig. 6g). In both mice and humans, RNA velocity analysis from single-cell RNA-seq indicated that *Ehf* expression progressed from CCR7⁻ cDC2s to CCR7int DCs (Cluster 1) and then to CCR7hi DCs (Cluster 2 and 3) (Fig. 6h). In humans, we

showed that *EHF* expression emerged in CCR7hi DCs after TLR7/8/9 stimulation.

## Discussion
In this manuscript, we present a transcriptional program that regulates maturation and immunosuppression of CCR7hi DCs. First, we demonstrated the physiological importance of EHF in DCs via in vivo infection, autoimmune and tumor models. Combining the results from conditional knockout mice, CUT&TAG analysis and overexpression studies, we found that EHF may directly promote the transcription of inhibitory molecules such as CCR7, CD200 and PD-L1. In addition, EHF suppressed the expression of the DC lineage-specific transcription factor IRF4 and promoted the expression of the inhibitory NFκB family members Rel.

Our results are consistent with previous literature regarding regulatory DCs. First, a pioneering study suggested that migratory cDCs in the lymph nodes may be more tolerogenic[24]. Tolerogenic cDC1s in the thymus and TLR-induced mature DCs were found to share similar expression profiles[23]. We found that TLR7/8/9 stimulation is a key requirement for the initiation of EHF. Second, highly immunosuppressive mature regulatory DCs (mregDCs) were identified among CCR7⁺ cDC1s and cDC2s from tumor-associated tissue[13]. We found that EHF was enriched in CCR7hi cDC1s and cDC2s, regulated similar inhibitory markers such as CD200 and PD-L1, and suppressed the anti-tumor T-cell response. Third, in the intestinal microenvironment, cDC1s were found to promote cross-tolerance to epithelial-derived antigens[28,30]. We found that EHF could suppress immunity in a DSS-induced colitis model. Fourth, in neonatal mice, cDC1s were found to exert MyD88-dependent tolerogenic effects during a temporal wave of apoptosis in developing lungs[29]. We found that EHF in cDCs is upregulated by apoptotic cells, and TLR7/8/9 upregulation of EHF is dependent on MyD88. Interestingly, according to scRNA-seq cluster analysis, cDC1s was absent after CpG-B stimulation. Instead, a population of XCR1int DCs emerged (Supplementary Information, Gating Strategy), which may be of similar origin to the previously reported regulatory CD103int DCs found in neonatal mice[29]. Since we showed that EHF repressed cDC1/2 lineage transcription factors IRF4, it is possible that high expression of EHF lead to a loss of strict lineage commitment to cDC1 or cDC2s.

TLRs play essential roles in regulating innate and adaptive immune responses[60]. In particular, the sensing of endogenous RNA or DNA by TLR7 and TLR9 is crucial for the autoactivation of dendritic cells and autoreactive B cells in a variety of autoimmune diseases, such as SLE and psoriasis[61,62]. TLR7 deficiency ameliorated symptoms in murine SLE models[63]. However, overexpression of TLR9 did not cause lupus-like autoimmunity[61,63,64]. Surprisingly, TLR9 seems to protect against SLE, even though it is required for the production of autoantibodies recognizing double-stranded DNA-associated antigens[61,65]. In contrast to TLR7 deficiency, TLR9 deficiency exacerbated SLE symptoms, suggesting that TLR9 somehow confers protection[61,65]. Recently, TLR9 has been shown to exacerbate SLE pathogenesis via an MyD88-dependent pathway, while simultaneously protect against SLE

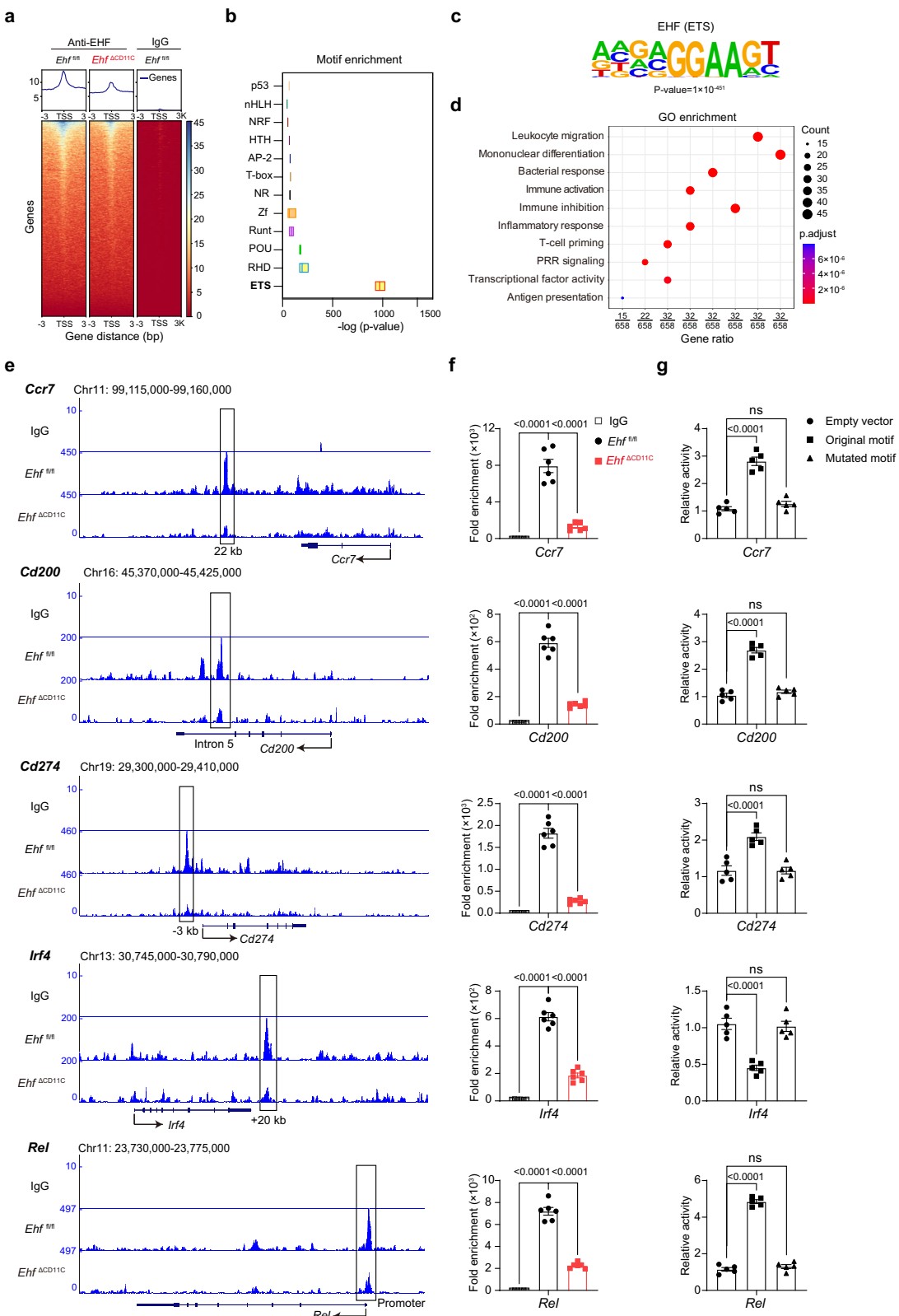

via an MyD88-independent pathway[66]. However, this study was conducted in B cells, and it remains unclear whether the same mechanism exists for DCs. In contrast, poly(I:C) is known to induce cDC1 expansion and an antitumor response in cancer[67]. The viral triggering of the TLR3 pathway may explain the lack of phenotype in virally-infected EHF-deficient mice. Further experiments are needed to determine whether the regulation of EHF by TLR3 is mediated by the TRIF-dependent

pathway. Finally, it unclear why EHF represses some genes while promotes other genes simultaneously. EHF could be associated with unknown co-activators or co-repressors on different gene loci.

A phenotype in EHF-deficient mice may not be observed if there is abundant early production of danger signals such as TLR3 signaling, GM-CSF and IFNγ. The transient and rapid down-regulation of EHF makes it difficult to ascertain its direct targets if

**Fig. 5 | CUT&TAG analysis suggests that EHF may directly regulate gene transcription in DCs. a** Genome-wide comparison of CUT&TAG signals at - 3 kb to 3 kb from the transcription start site (TSS) in the indicated samples was performed by deepTools. **b** Motif enrichment analysis was obtained with HOMER. Minima and maxima represent the values within 1.5 × interquartile range (IQR) of the box bounds; the center line denotes the median (50th percentile); the bounds of the box correspond to the 25th (lower) and 75th (upper) percentiles; There are no whiskers. Statistical analysis was calculated using the two-tailed hypergeometric test, with Benjamini−Hochberg false discovery rate (FDR) adjustment for multiple comparisons. The log($p$-value) reflects the statistical significance of transcription factor motif enrichment for each family. **c** The de novo EHF motif was calculated by HOMER. **d** GO analysis of enriched CUT&TAG signals was obtained with Cluster-Profiler. Gene ratio indicates the proportion of enriched genes, bubble size represents enriched gene count, and bubble color reflects the adjusted $P$-value (p.adjust). Enrichment significance was determined by the two-tailed

hypergeometric test with FDR correction for multiple comparisons. **e** CUT&TAG profiling of the DNA-binding sites by EHF around *Ccr7*, *Cd200*, *Cd274*, *Irf4* and *Rel* loci was visualized with the UCSC genome browser. **f** Primers were designed for a randomly chosen ETS motif in the boxed area in indicated genes, and qPCR were performed on genomic DNA pulled from CUT&TAG ($n = 6$ independent runs). **g** The 293 T cells were transfected with either vector pMXs-mock or pMXs-EHF, in conjunction with luciferase reporter plasmids with indicated ETS motifs or mutated motifs. Results are expressed as fold induction relative to that of the cells transfected with the mock vector after normalization of firefly luciferase activity according to Renilla luciferase activity ($n = 5$ independent replicates). $P$-values were calculated by one-way ANOVA with Dunnett's multiple comparisons test (ns, $P > 0.05$; *, $P < 0.05$; **, $P < 0.01$; ***, $P < 0.001$; ****, $P < 0.0001$). All experiments were repeated three times independently. All the data are presented as the means ± SEMs and are representative of three independent experiments.

EHF is absent. If large amounts of GM-CSF and IFNγ are present, EHF will be quickly downregulated or not expressed at all in DCs. This hypothesis is supported by the complete lack of EHF expression in GM-CSF-cultured moDCs both in vitro and in vivo, even with CpG-B stimulation. In the B16-OVA melanoma model, CTL responses were enhanced in Ehf-deficient mice, suggesting that EHF in DCs may play a critical role in regulating CTL function in vivo. In viral and bacterial infection models, there were some limited phenotypes on CTLs. However, in the in vitro co-culture experiments, DCs lacking EHF did not affect the development or activation of CTLs. There could be two reasons for this discrepancy. First, we found that EHF in DCs regulates Th1 response, which directly controls CTL response. However, this CTL-Th1 interplay is difficult to capture in an in vitro CTL assay with just CD8⁺ T cells. Second, we found that EHF is downregulated by IFNγ. It is well known that the IFNγ production by CTLs dwarfs that of Th1s. Therefore, the microenvironmental IFNγ concentration may accumulate more in a cell culture dish than what happens in vivo. Thus, EHF in DCs may get downregulated more quickly or may not get even upregulated in a CTL assay compared to an Th1 assay. A deficiency in EHF upregulation may result in a lack of phenotype in vitro. This dynamic negative regulation could also explain the very limited CTL phenotype in infection models, in which IFNγ is massively produced compared to the tumor model.

In tumors, the enrichment of a CCR7⁺ DC signature is associated with improved survival in multiple cancers[68]. First, this discrepancy might be the different clusters of CCR7⁺ DC. EHF is not expressed in CCR7ˡᵒ (Cluster 1), which is still considered to be CCR7⁺. This CCR7⁺ population in cancers might contains more CCR7ˡᵒ (Cluster 1). Second, EHF expression is downregulated quickly by GM-CSF and IFNγ in CCR7⁺ DCs after maturation. This CCR7⁺ population in the more survivable "hot" cancers may not have sustained expression of EHF due to the abundance of such cytokines.

There was a selective inhibitory role for EHF in Th1/Th17 development. However, we examined various cytokine production by EHF-deficient DCs and found no specific pattern for Th1/Th17 related molecules. EHF seems to be broadly immunosuppressive in terms of its downstream targets. A possible mechanism could be the sustained up-regulation of EHF expression in type I and type III immune responses. We found that TLR7/8/9 activation via apoptotic cells is a main trigger of EHF upregulation. TLR7/8/9 signaling are well-known to induce Th1/Th17 response, and the increased apoptosis in type I and type III immune response may reinforce EHF expression. Therefore, the differential response in Th1/Th17 response could be due to a longer activation of TLR7/8/9 and subsequent sustained EHF expression in DCs.

In summary, our study identified EHF as an immunosuppressive transcription factor that may override classical inflammatory programming in mature CCR7⁺ DCs. Second, we highlighted a conserved tolerogenic DC response to TLR7/8/9. Cell-based therapies that modulate EHF expression via lentiviruses, RNAi or CRISPR/Cas9 technology

could be used to potentially treat autoimmune diseases, cancer and pathogenic infections.

## Methods

### Transgenic mice

All mice were bred and maintained under specific pathogen-free conditions at Shanghai Model Organisms Center Inc., and BSL3 facilities at Sun Yat-sen University, according to the institutional guidelines and protocols approved by the Animal Ethics Committee of Sun Yat-sen University, Guangzhou, Guangdong, China. C57BL/6 J (JAX: 000664, B6 or WT), B6.Cg-Tg (TcraTcrb) 425Cbn/J (JAX: 004194, OT-II), B6.SJL-PtprcᵃPepcᵇ/BoyJ (JAX: 002014, CD45.1⁺), B6.Cg6-Tg (TcraTcrb) 1100Mjb/J (JAX: 003831, OT-I), and B6.Cg-Tg (*Itgax-Cre*) 1-1Reiz/J (JAX: 008068, Itgax-Cre) mice were purchased from the Jackson Laboratory. C57BL/6Smoc-*Myd88* ᵉᵐ¹ˢᵐᵒᶜ (#NM-KO-190192, *Myd88* ⁻/⁻) mice were purchased from Shanghai Model Organisms Center Inc. Age- and sex-matched mice aged 6–16 weeks were used in this study. Sex was not considered in the study design and analysis due to the limited numbers of available mice.

*Ehf* ᶠˡ/ᶠˡ mice on the B6 background were generated by Shanghai Model Organisms Center Inc. The genetic modification strategy and subsequent validation are described in Supplementary Fig. 2. The mice were genotyped via PCR (*Ehf*-fl-Fwd: TTGTCTCCTTGTCCGCATCC; *Ehf*-fl Rvs: CTGAATACCCACGGTGTGCT). The insertion of loxP sites was confirmed by the expected 402 bp PCR band, as the wild-type band was 338 bp in length. Cell-specific deletion of the *Ehf* allele was achieved after Cre-mediated recombination by crossing with Itgax-Cre.

### In vivo infection models

For lethal *L. monocytogenes* infection, LM-OVA was grown in TSB (HKM, #024051) medium to an OD₆₀₀ of approximately 0.25, diluted in PBS and intravenously injected ($2 × 10^4$ CFU) in a volume of 0.2 ml per mouse. Survival was monitored daily. For antigen-specific T-cell response, mice were additionally injected intravenously with $1 × 10^5$ OT-I prior to infection of LM-OVA ($8 × 10^3$ CFU). After 5 days, the mice were analyzed.

For VSV infection, $1 × 10^6$ PFU were diluted in PBS and injected intravenously. The spleens was collected to evaluate viral titers via qPCR.

For *C. rodentium* infection, *Citrobacter rodentium* strain DBS100 (ATCC, #51459) was grown in LB medium, diluted in PBS and orally inoculated with $2 × 10^9$ CFU in a total volume of 200 μL per mouse.

For all pathogenic infections, we adhered to the early end point by institutional ethical board: the experiments will be terminated and animals sacrificed by cervical dislocation if body weight loss reached ≥ 20% from baseline, the clinical severity score reached its maximum, or moribund condition/severe clinical signs (e.g., persistent anorexia/ad libitum water refusal, severe dehydration, dyspnea, neurological symptoms) occurred.

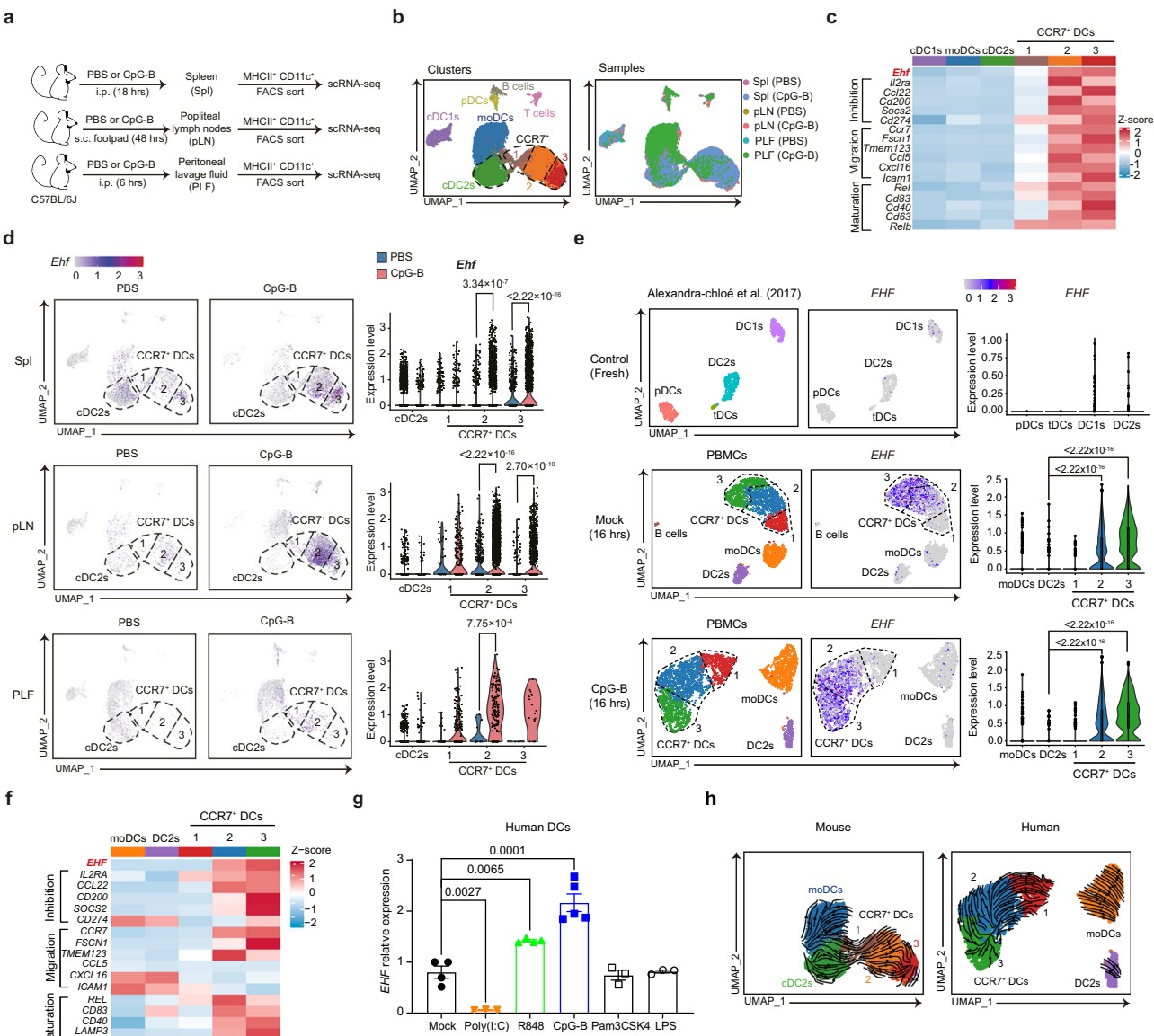

**Fig. 6 | EHF is expressed by human and murine CCR7ʰⁱ DCs. a** Experimental design for the scRNA-seq analysis of murine DCs from different organs. FACS-sorted MHCII⁺CD11c⁺ cells from B6 spleens (Spl), popliteal lymph nodes (pLN), and peritoneal lavage fluid (PLF) after PBS or CpG-B injection were subjected to 10 × Genomics Chromium single-cell RNA sequencing. **b** After integrated transcriptomic analysis of MHCII⁺CD11c⁺ cells, UMAP dimensionality reduction on the integrated dataset (left) and UMAP colored by different samples (right) are shown. **c** Heatmap depicting the average Z score of selected differentially expressed genes is shown with genes belonging to the CCR7ʰⁱ DC signature set as the standard. **d** Feature plots (left) and violin plots (right) depicting *Ehf* expression in all clusters from different organs with or without CpG-B injection. The plots are based on an integrated analysis of all six samples. **e** UMAPs of FACS-sorted CD1C⁺/CD11C⁺HLA-DR⁺ human DCs from untreated (top, GSE94820) or PBS-treated (middle), and CpG-B

stimulated (bottom) cells are shown on the left. Feature plots and violin plots depicting *EHF* expression are shown on the right. **f** Heatmap depicting the average Z score of selected genes from integrated human DCs (PBS and CpG-B-treated) is shown with genes belonging to the CCR7ʰⁱ DC signature set as the standard. **g** Human DCs were FACS-sorted from PBMCs of healthy volunteers, and *EHF* transcript levels were measured via qPCR after in vitro stimulation with the indicated TLR ligands for 14 hrs (Mock, $n = 4$; Poly(I:C), $n = 3$; R848, $n = 4$; CpG-B, $n = 5$; Pam3CSK4, $n = 3$; LPS, $n = 3$ individuals). **h** RNA velocity analysis of EHF expression in murine (left) and human (right) DCs is shown. *P*-values were calculated by two-sided Wilcoxon rank sum test (**d**, **e**) or one-way ANOVA with Dunnett's multiple comparisons test (**g**) (ns, $P > 0.05$; *, $P < 0.05$; **, $P < 0.01$; ***, $P < 0.001$; ****, $P < 0.0001$). All the data are presented as the means ± SEMs and are from one independent experiment.

For tumor models, the melanoma cell line B16 (ATCC, CRL-6475) was stably transfected with OVA-expressing plasmids in house and cultured at 37 °C in RP10. Next, $5 \times 10^5$ tumor cells were injected subcutaneously (s.c.) in 100 µL of PBS in each flank. On Day 11, the mice were intravenously injected with $1 \times 10^5$ OT-I cells. Three days after OT-I injection, the mice were analyzed. We adhered to the early end point by institutional ethical board: experiments will be terminated and animals sacrificed by cervical dislocation if tumor size exceeded 15 mm in diameter or 1.8 cm³ in volume (the institutional ethics maximum

tumor size), or if the animals presented with a body condition score < 2.0, reduced activity, failure to respond to gentle stimulation, lethargy, piloerection, persistent hunched posture, or tumor ulceration. Tumor size was determined by the formula: length × width × width × 0.5 by a digital caliper.

For the DSS-based colitis model, 7 to 8-week-old sex-matched cohoused littermates were administered 2.5% DSS (36–50 kDa; MP Biomedicals, #160110) in their drinking water for consecutive days. Survival and clinical parameters, such as weight loss, rectal

bleeding and diarrhea, were monitored daily. During the duration of the experiment, the disease activity index (DAI) score was determined to evaluate the clinical progression of colitis. The DAI is the combined score of weight loss compared to initial weight, stool consistency, and bleeding. Scores were defined as follows: weight loss: 0 (no loss), 1 (1–5%), 2 (5–10%), 3 (10–20%), and 4 (>20%); stool consistency: 0 (normal), 2 (loose stool), and 4 (diarrhea); and bleeding: 0 (no blood), 1 (hemoccult positive), 2 (hemoccult positive and visual pellet bleeding), and 4 (gross bleeding, blood around anus). The DAI was determined daily during DSS treatment. For assessment of intestinal pathology, an expert blinded to the genotypes scored H&E-stained intestinal sections based on three parameters of tissue damage and four parameters of inflammation, and multiplied these scores by a factor accounting for the extent of the tissue being affected, as described previously. Formalin-fixed and paraffin-embedded intestinal tissue was sectioned and stained with hematoxylin and eosin. A semiquantitative composite scoring system, computed as the sum of five histological subscores, multiplied by a factor based on the extent of inflammation, was used for the assessment of spontaneous intestinal inflammation. Histological subscores were as follows (for each parameter: 0, absent; 1, mild; 2, moderate; 3, severe): mononuclear cell infiltrate (0–3), crypt hyperplasia (0–3), epithelial injury/erosion (0–3), polymorphonuclear cell infiltrates (0–3) and transmural inflammation (0, absent; 1, submucosal; 2, one focus extending into muscularis and serosa; 3 up to five foci extending into muscularis and serosa; 4, diffuse). The extent factor was derived according to the fraction of bowel length involved in inflammation: 1, <10%; 2, 10–25%; 3, 25–50%; and 4, >50%. No spontaneous colonic inflammation was detected in any of the reported genotypes. We adhered to the early end point by institutional ethical board: experiments will be terminated and animals sacrificed by cervical dislocation if body weight loss reached ≥ 20% from baseline, severe systemic signs or complications (e.g., persistent anorexia, hypothermia, dyspnea, anemia, moribund state), maximum DAI score as defined in the protocol, or other critical conditions (e.g., suspected intestinal perforation, severe rectal prolapse, severe perianal infection) were observed.

## Flow cytometry

Cell suspensions of bone marrow and splenocytes were subjected to red blood cell lysis. The cells were subsequently washed and resuspended in FACS buffer (1% FCS and 0.1% sodium azide in PBS). Human PBMCs were isolated via Ficoll (Solarbio, #P4350) gradient centrifugation. The cells were stained with the indicated cell surface markers with an Fc blocker (BD, #553141) to exclude nonspecific staining. For intracellular staining, the cells were surface stained, subsequently fixed and permeabilized using an intracellular staining kit (eBioscience, #88-8824-00). For transcription factor staining, the cells were surface stained, subsequently fixed and permeabilized using a transcription factor intracellular staining kit (eBioscience, #00-5523-00) before staining. Flow cytometry was performed on a Beckman Colter CytoFLEX and analyzed using FlowJo software (Tree Star, version X). The MFI was calculated according to the genomic mean by the FlowJo software.

To measure cytokine production, cDCs, pDCs, macrophages were plated in complete RPMI-1640 (RPMI-1640 with 10% FCS, 1% L-glutamate, 1% penicillin/streptomycin, and 55 μM 2-mercaptoethanol), and stimulated with 1 μg/mL Pam3CSK4 (InvivoGen, #tlrl-pms), 100 ng/mL LPS (Sigma-Aldrich, #L2654-1MG), 500 ng/mL poly(I:C) (InvivoGen, #tlrl-picw), 1 μg/mL R848 (InvivoGen, #tlrl-r848), 100 nM CpG-B (InvivoGen, #tlrl-1668) for 2 h (TNF, IL-6, IL-12p40) or 4 h (IFNα), then added brefeldin A (eBioscience, # 00-4506-51) for 10 h (TNF, IL-6, IL-12p40) or 5 hrs (IFNα) before intracellular staining. Anti-mouse TNF (Invitrogen, #25-7321-82), IL-6 (Invitrogen, #48-7061-82) and IL-12p40

(Biolegend, #505206) were used at dilutions of 1:400, anti-mouse IFNα (Pbl assay science, clone: RMMA-1, #22100-3) were used at dilutions of 1:100. The antibody information for flow cytometry is shown in Supplementary Table 1.

## In vivo and in vitro T-cell priming

For the Th1 response, all mice were injected intravenously with $1 \times 10^5$ OT-II cells before immunization. Mice were immunized subcutaneously (s.c.) in the rear footpads with 50 μg of OVA (Sigma, #A5503) and 20 μg of CpG-B in 40 μL of PBS. Popliteal lymph nodes and spleens were collected at the indicated hours for FACS analysis.

For the Th2 response, all mice were injected intravenously with $1 \times 10^5$ OT-II cells before immunization. Mice were immunized subcutaneously (s.c.) in the rear footpads with 50 μg of OVA (Sigma, #A5503), 50 μg of papain (Sigma, #76216) and 20 μL IFA (Sigma, #F5506) in 20 μL of PBS. Popliteal lymph nodes were collected at the indicated hours for FACS analysis.

For the Tfh response, all mice were injected intravenously with $1 \times 10^5$ OT-II cells before immunization. Mice were immunized intraperitoneally with $5 \times 10^8$ SRBCs (Hqbio, # HQ80073-004) conjugated with OVA (Sigma, #A5503) in 100 μL of PBS. For conjugation of OVA to SRBCs, SRBCs were washed with PBS three times, incubated with 4 mL of 30 mg/ mL ice-cold OVA in PBS and crosslinked with 1 mL of 100 mg/ mL EDCI (Sigma, #E7750) for 1 hr on ice with occasional mixing, followed by washing four times in PBS to remove the free OVA. Spleens were collected at the indicated time for FACS analysis.

For the antigen cross-presentation assay with cell-associated OVA, all mice were injected intravenously with $1 \times 10^6$ OT-I cells before immunization. BALB/c splenocytes were loaded with OVA by osmotic shock. Cells were incubated in hypertonic medium (0.5 M sucrose, 10% polyethylene glycol, and 10 mM Hepes in RPMI 1640, pH 7.2) containing 10 mg/ml OVA for 10 min at 37 °C, and then prewarmed hypotonic medium (40% H2O and 60% RPMI 1640) was added for an additional 2 min at 37 °C. After washing and irradiation (1350 rad), $2 \times 10^6$ OVA-loaded cells were injected intravenously into mice. Spleens were collected at the indicated time for FACS analysis.

For in vitro T cell stimulation, naive CD44$^{lo}$CD62L$^{hi}$ OT-II or CD44$^{lo}$CD62L$^{hi}$ OT-I cells were sorted with a BD FACS Aria III flow cytometer. DC subsets were FACS-purified after enrichment with streptavidin microbeads. DCs and T cells were co-cultured at a ratio of $2 \times 10^4$ DCs to $1 \times 10^5$ T cells in the presence of 100 nM CpG-B (InvivoGen, #tlrl-1668) and 1 μg/mL OVA$_{323-339}$ (MCE, # HY-P0286) or 200 nM SIINFEKL peptide (GenScript, #RP20398), under Th0 conditions (PBS) or with the addition of polarizing cytokines (Th1: 50 ng/ml IL-12p70 (PeproTech, #210-12); Th2: 50 ng/ml IL-4 (PeproTech, #214-14); Th17: 20 ng/ml TGF-β1 (Novoprotein, #C16W), 20 ng/ml IL-6 (PeproTech, #216-16), 10 ng/ml IL-1β (PeproTech, #211-11B); iTreg: 2 ng/ml TGF-β1 (Novoprotein, #C16W), 50 ng/ml IL-2 (PeproTech, #212-12). 5 ng/ml IL-2 (PeproTech, #212-12) was added to Th0, Th1 and iTreg on Day 3. Cytokine production was assessed after 5 days of culturing.

To measure cytokine production, T cells were plated in complete RPMI-1640, and stimulated with cell stimulation cocktail (Invitrogen, #00-4970-03), 1 μg/mL OVA$_{323-339}$ peptide (MCE, #HY-P0286) or 200 nM SIINFEKL peptide (GenScript, #RP20398) for 6 hrs before intracellular cytokine staining.

## Cell isolation

For harvesting immune cells from organs, spleens, lymph nodes and tumors were digested in collagenase I (Sigma-Aldrich, #C0130) and DNase I (Roche, #10104159001) for 1 hr at 37 °C with stirring in RPMI, then organs were minced, ground, and passed through a 70 μm nylon mesh. For the colon, colons were cut longitudinally and laterally into small pieces. The tissues were incubated with epithelial removal buffer (20 mM EDTA, 1 mM sodium pyruvate, 10% FCS and 10 mM HEPES in PBS) for 30 min at 37 °C, then washed with cold PBS,

after which the epithelial cells were removed with scissors. The remaining tissue were digested with collagenase D (Roche, # COLLD-RO) and DNase I (Roche, #10104159001) for 1 hr at 37 °C with stirring in RPMI. After enzymatic treatment, colon tissues were further dissociated through a 70 μm nylon cell strainer. For the isolation of lymphocytes from the tumor and colon, single-cell suspensions were then separated using a 30%/70% Percoll (Biosharp, #BS012) density gradient. Red blood cells were lysed with cold ACK buffer (150 mM ammonium chloride, 10 mM potassium bicarbonate, and 0.1 mM EDTA in ddH₂O). The cells were counted using a Beckman Colter CytoFLEX flow cytometer. Before cell sorting, cDCs and pDCs were stained with biotinylated anti-CD11c (eBioscience, #13-0114-82) at a 1:500 dilution for 20 min at 4 °C. Then, the cells were washed 3 times with MACS buffer (10 mM EDTA, 10% FCS and 10 mM HEPES in PBS) and labeled with streptavidin microbeads (Miltenyi Biotech, #130-048-101) for 15 min at 4 °C. Target cells were then purified by positive selection using an LS column in a QuadroMACS Separator. Then, the bead-purified cDCs were sorted as MHCII⁺ CD11c⁺ cells, and the bead-purified pDCs were sorted as B220⁺SiglecH⁺BST-2⁺ cells (purity>95%). Cell-sorting experiments were conducted on a BD FACS Aria III flow cytometer.

### In vivo and in vitro DC assays
For CpG-B stimulated cDCs in the spleen, mice were injected intraperitoneally (i.p.) with or without 60 μg of CpG-B in 100 μL of PBS. Spleens were harvested at the indicated time for FACS analysis and qPCR.

For CpG-B stimulated cDCs in the lymph nodes, mice were injected subcutaneously in the rear footpads with or without 10 μg of CpG-B in 40 μL of PBS. Popliteal lymph nodes were collected at the indicated time for FACS analysis and qPCR.

For GM-CSF and IFNγ blocking in the spleen, mice were injected intraperitoneally (i.p.) with or without 100 μg anti-GM-CSF (Invitrogen, #MA5-23799), 100 μg anti-IFNγ (Invitrogen, #16-7311-85) or 60 μg CpG-B in 100 μL of PBS. Spleens were harvested at 14 h for qPCR.

For in vitro stimulation, FACS-sorted DC subsets were plated in complete RPMI-1640 (RPMI-1640 with 10% FCS, 1% L-glutamate, 1% penicillin/streptomycin, and 55 μM 2-mercaptoethanol), and stimulated with 1 μg/mL Pam3CSK4 (InvivoGen, #tlrl-pms), 100 ng/mL LPS (Sigma-Aldrich, #L2654-1MG), 500 ng/mL poly(I:C) (InvivoGen, #tlrl-picw), 1 μg/mL R848 (InvivoGen, #tlrl-r848), 100 nM CpG-B (InvivoGen, #tlrl-1668), 1 μg/mL mAb GM-CSF (Invitrogen, #MA5-23799) and the indicated cytokines for 9–12 h (FACS analysis) or 14 hrs (qPCR).

For cytochalasin D treated cDCs, FACS-sorted cDCs were resuspended in complete RPMI-1640 (RPMI-1640 with 10% FCS, 1% L-glutamate, 1% penicillin/streptomycin, and 55 μM 2-mercaptoethanol), and treated with 10 μg/mL cytochalasin D (Sigma, #C2618) for 30 min at 37 °C, washed three times with medium, and then cDCs were plated in complete RPMI-1640 medium and stimulated with 100 nM CpG-B (InvivoGen, #tlrl-1668) for 14 h. The cytokine information for DC assays is shown in Supplementary Table 2.

### Migration assay
The migratory capacity of CCR7⁺ cDCs in a CCR7-dependent manner was measured by a chemotaxis assay as previously described[69,70]. Briefly, splenic cDCs (~10%) were purified with CD11c-biotin and MACS selection (5 × 10⁵, 100 μL), then seeded in the top chambers with 24-well plate with 8 μm pore size (Corning Costar, #3422). Lower chamber wells contained 500 μL of medium supplied with different dose (0.5, 5.0, 20 ng/mL) of CCL19 (PeproTech, #250-27B) and CCL21 (Pepro-Tech, #250-13) or 20 ng/mL CCL2. The plates were incubated for 12 h at 37 °C at 5% CO₂. Then, the top chambers were removed and the migrated cells in the bottom chamber were collected and acquired on a Beckman Colter CytoFLEX for cell count.

### Primary cell culture
cDC^FL-Notch were cultured as previously described[54]. In brief, 4 × 10⁶ bone marrow cells per well were cultured in tissue culture-treated 24-well plates in 2 mL of complete RPMI-1640 (RPMI-1640 supplemented with 10% FCS, 1% L-glutamate, 1% sodium pyruvate, 1% MEM with nonessential amino acids, 1% penicillin/streptomycin, and 55 μM 2-mercaptoethanol), supplemented with 100 ng/mL Flt3L ((PeproTech, #250-31 L). On Day 3 of differentiation, half of the volume of cells in DC medium from each well was transferred to a single well containing a monolayer of mitomycin-treated OP9-Notch2 cells in 24-well plates. On Day 6, the cells were stimulated with 100 nM CpG-B (InvivoGen, #tlrl-1668) for 12–14 h for FACS analysis. cDC^FL-Notch were gated as live⁺ B220⁻MHCII⁺CD11c⁺ cells.

For BMDCs, 10 × 10⁶ bone marrow cells per well were cultured in tissue-culture-treated 6-well plates in 4 mL of RPMI-1640 supplemented with 10% FCS, 1% L-glutamate, 1% sodium pyruvate, 1% MEM with nonessential amino acids, 1% penicillin/streptomycin, and 55 μM 2-mercaptoethanol, and 20 ng/mL GM-CSF (PeproTech, #315-03). Half of the medium was removed on Day 2, and new medium supplemented with 40 ng/mL GM-CSF and warmed to 37 °C was added. The culture medium was entirely discarded on Day 3 and replaced with fresh warm medium containing 20 ng/mL GM-CSF. On Day 5, the cells were stimulated with 100 nM CpG-B (InvivoGen, #tlrl-1668) for 12–14 h for FACS analysis. BMDCs were gated as live⁺ MHCII⁺CD11c⁺ cells.

For retroviral transduction of primary cells, Phoenix packaging cells (ATCC, #CRL-3214) were transfected with 20 μg of the plasmid pCL-Eco (Addgene, #25099) and 16 μg of pMXs-EHF, or the empty vector pMXs, with 90 μL of the transfecting reagent polyethylenimine (Polysciences, #23966-1) at 1 mg/mL in a 10 cm culture dish. After 48 h, the supernatant containing retrovirus was harvested and filtered. On Days 1 and 2, 2 ml of retroviral supernatant was added, and the cells were spin-infected at 2500 rpm for 90 min at room temperature. On Day 3, fresh medium was added to the cells, and they were cultured for additional days. All retroviral transduction rates were measured by GFP via FACS.

### RNA extraction and quantitative PCR
TRIzol reagent (GenStar, P118-05) and chloroform were added to homogenize single cells, after which RNA precipitation, washing and resuspension were performed following the manufacturer's protocol. The extracted RNA was reverse transcribed according to the manufacturer's protocol (HiScript III RT SuperMix for qPCR, Vazyme, R323-01). Quantitative RT-PCR analysis was performed with SYBR Select Master Mix (GenStar, A301-10) using StepOne Plus (Life Sciences). All the data were normalized to β-actin expression. The primers used are shown in Supplementary Table 3.

### Generation of an anti-EHF monoclonal antibody
Mice were immunized with a partial recombinant EHF antigen (1–127AA) and adjuvants for 3 rounds. After immunization, antibody-producing cells were harvested from the mice and fused with tumor cells to become hybridomas. The hybridomas were screened for positive monoclonal antibodies via Western blotting. One clone (XL-32) was chosen for subsequent experiments. The anti-EHF monoclonal antibody was generated with assistance from AtaGenix Laboratories Co., Ltd. (Wuhan), China. This antibody is generated in-house and would be available to the public with any reasonable request.

### Western blotting
For Western blotting, protein samples were subjected to SDS-PAGE on a 4–12% gradient gel and then transferred to a membrane. The membranes were blocked with 5% dry milk and incubated with primary antibodies overnight. After washes with PBS containing Tween 20, the membranes were incubated with secondary antibodies for 1.5 h. The

bands were visualized by chemical composition using ChemiDoc Touch (Bio-Rad).

The following antibodies were used for Western blotting at 1:5000 dilution:

Tubulin (Abcam, host: mouse, #ab78078)
β-actin (Abcam, host: mouse, #ab6276)
Anti-mouse IgG H&L (HRP) (Abcam, host: goat, #ab6789).

## CUT&TAG

For cDC$^{FL-Notch}$ samples, cDC$^{FL-Notch}$ were cultured for 6 days and stimulated with 100 nM CpG-B (InvivoGen, #tlrl-1668). Live B220$^-$MHCII$^+$CD11c$^+$ cells were subjected to FACS sorting after 18 hrs of CpG-B stimulation for CUT&TAG library preparation.

For spleen samples, mice were immunized intraperitoneally (i.p.) with or without 60 μg CpG-B in 100 μL of PBS. Splenic cells were collected 18 h after CpG-B injection for cDC1 (live MHCII$^+$CD11c$^+$ XCR1$^+$CD8α$^+$CD11b$^-$) and cDC2 (live MHCII$^+$CD11c$^+$CD11b$^+$XCR1$^-$CD8α$^-$) FACS-sorting for CUT&TAG library preparation.

cDC$^{FL-Notch}$ were cultured for 6 days and stimulated with 100 nM CpG-B (InvivoGen, #tlrl-1668). Live B220$^-$MHCII$^+$CD11c$^+$ cells were subjected to FACS sorting after 18 hrs of CpG-B stimulation. For CUT&TAG library preparation, the CUT&TAG assay was performed following the manufacturer's instructions (Vazyme, #TD903). In brief, the cells were harvested, counted and centrifuged for 5 min at 600 × g at room temperature. A total of 500,000 cells per sample were washed twice in 1.5 mL of wash buffer by gentle pipetting. Ten microliters of activated conA beads were added to each sample (100 μL aliquots of cells) and incubated at RT for 10 min. The unbound supernatant was removed, and the bead-bound cells were resuspended in 50 μL of antibody buffer containing a 1:50 dilution of the appropriate primary antibody. The primary antibody incubation was performed overnight at 4 °C. The primary antibody was removed by placing the tube on the magnet stand to clear and removing all of the liquid. An appropriate secondary antibody (goat anti-mouse IgG antibody for a mouse primary antibody) was diluted 1:100 in 50 μL of Dig-Wash buffer, and the cells were incubated at RT for 45 min. The cells were washed using the magnet stand 3 times for 5 min in 0.2 mL Dig-Wash buffer to remove unbound antibodies. A 1:200 dilution of the pA-Tn5 adapter complex (~0.04 μM) was prepared in Dig-300 Buffer. After the liquid was removed on the magnet stand, 100 μL was added to the cells with gentle vortexing, followed by incubation with pA-Tn5 at RT for 1 hr. The cells were subsequently washed 3 times for 5 min in 0.2 mL of Dig-300 Buffer to remove unbound pA-Tn5 protein. Next, the cells were resuspended in 50 μL of tag mentation buffer (40 μL of Dig-300 Buffer and 10 μL of 5 × TTBL) and incubated at 37 °C for 1 hr. For the extraction of the DNA, 5 μL of proteinase K, 100 μL of buffer L/B and 20 μL of DNA extract beads was added to 100 μL of sample, which was incubated for 10 min at 55 °C. The tubes were placed on a magnet stand to clear, then the liquid was carefully removed. Without disturbing the beads, the beads were washed twice in 1 mL of 80% ethanol. After the samples were allowed to dry for ~5 min, 22 μL of ddH$_2$O was added, and the tubes were vortexed, quickly spun and allowed to sit for 5 min. The tubes were placed on a magnet stand, and the liquid was withdrawn to a fresh tube.

For the amplification of the libraries, 15 μL of DNA was mixed with 5 μL of a universal i5 and a uniquely barcoded i7 primer (Vazyme, #TD202), using a different barcode for each sample. A volume of 25 μL of 2 × CAM was added, and the solution was mixed. The sample was placed in a thermocycler with a heated lid using the following cycling conditions: 72 °C for 3 min (gap filling); 95 °C for 3 min; 15 cycles of 98 °C for 10 s and 60 °C for 5 s; final extension at 72 °C for 1 min and a hold at 4 °C. Post-PCR clean-up was performed by adding a 1.1 × volume of Ampure XP beads (Vazyme, #N411), and the libraries were incubated with the beads for 5 min at RT, washed twice gently in 80% ethanol, and eluted in 20 μL of ddH$_2$O. The size distribution of the libraries was determined by LabChip Touch, and the libraries were mixed to achieve equal representation as desired, aiming for a final concentration as recommended by the manufacturer. Paired-end Illumina sequencing (PE150) using Illumins HiSeq 3000 (Annoroad Gene Technology, Guangzhou, China) was performed on the barcoded libraries following the manufacturer's instructions. The raw FASTQ data were first processed through fastp version 0.23.2 with the options: -q 20 -u 50 -n 15 -l 50, and all the downstream analyses were based on high-quality clean data. An index of the reference genome was built, and paired-end clean reads were aligned to the reference genome (mm10) using Bowtie2 version 2.2.5 with default parameters. For peak calling, the parameters used were macs2 callpeak--shift 0 --extsize 200 --nomodel -B --SPMR -g mm -p 0.05. Peak assignment was performed on the intersected bed file using deepTools: computeMatrix reference-point -p 15 --referencePoint TSS -b 3000 -a 3000 -R ucsc_refseq.bed -S Ehf$^{fl/fl}$.bw Ehf$^{ΔCD11C}$.bw IgG.bw --skipZeros -out./ wt_cko.TSS.gz --outFileSortedRegions./wt_cko.genes.bed. Then, a heatmap was constructed with plotHeatmap. For identification of motifs enriched in peak regions over the background, HOMER's motif analysis (findMotifsGenome.pl), which included known default motifs and de novo motifs, was used with default parameters. The UCSC genome browser was used to visualize the CUT&TAG tracks.

Gene Ontology (GO) enrichment analysis of differentially expressed genes was implemented by the clusterProfiler R package (v4.2.2). GO terms with corrected P-value < 0.05 were considered significantly enriched by differential expressed genes. The ClusterProfiler R package was used to test the statistical enrichment of differential expression genes in GO pathways.

## Luciferase reporter assays

For luciferase analyses, 293 T cells were first transfected with pMXs-EHF or control vector (pMXs-vector), then they were transfected with the following vectors: pGL3-Ccr7-original motif, pGL3-Ccr7-mutated motif, pGL3-Cd200-original motif, pGL3-Cd200-mutated motif, pGL3-Cd274-original motif, pGL3-Cd274-mutated motif, pGL3-Irf4-original motif, pGL3-Irf4-mutated motif, pGL3-Rel-original motif, pGL3-Rel-mutated motif and pGL3-empty vector (pGL3-Promoter). Forty-eight hours later, cells were subjected to dual luciferase analyses with the Dual Luciferase Reporter Assay Kit (Vazyme, #DL101-01). The results were expressed as fold induction relative to the cells transfected with the control vector after normalization to renilla activity. ETS binding sites were mutated to GGGGGTTTTT. The ETS-motif-based primers used to validate CUT&TAG samples by real-time quantitative PCR are shown in Supplementary Table 3.

## Single-cell RNA-sequencing

For spleen samples, mice were immunized intraperitoneally (i.p.) with or without 60 μg CpG-B in 100 μL of PBS. Splenic cells were collected 18 hrs after CpG-B injection for cDC (live MHCII$^+$CD11c$^+$) FACS-sorting for scRNA-seq library preparation.

For pLN samples, mice were immunized subcutaneously (s.c.) in the rear footpads with or without 10 μg of CpG-B in 40 μL PBS. Popliteal lymph node cells were collected 48 h after CpG-B injection for cDC (live MHCII$^+$CD11c$^+$) FACS-sorting for scRNA-seq library preparation.

For PLF samples, mice were immunized intraperitoneally (i.p.) with or without 10 μg of CpG-B in 100 μL of PBS, and peritoneal cavity cells were obtained by lavage of the peritoneal cavity twice with 5 mL of ice-cold PBS. The cells were collected 6 h after CpG-B injection for cDC (live MHCII$^+$CD11c$^+$) FACS-sorting for scRNA-seq library preparation.

For pooled samples, mice were immunized intraperitoneally (i.p.) with or without 60 μg CpG-B in 100 μL of PBS. Splenic cells were collected 12 h after CpG-B injection. cDCs (live MHCII$^+$CD11c$^+$), pDCs (live SiglecH$^+$BST-2$^+$), macrophages (live F4/80$^+$CD11b$^+$), monocytes (live CD11b$^+$Ly6C$^{hi}$Ly6G$^{int}$), neutrophils (live CD11b$^+$Ly6C$^{int}$Ly6G$^{hi}$), T cells

(live CD3ε⁺TCR-β⁺), B cells (live CD19⁺B220⁺) were FACS-sorted at roughly equal cell numbers and pooled for scRNA-seq library preparation.

For scRNA-seq library preparation, single cells were encapsulated in emulsion droplets using a Chromium Controller (10 × Genomics). The scRNA-seq libraries were constructed using the Chromium Single Cell 3′ Library and Gel Bead & Multiplex Kit (10 × Genomics, V3.1) following the manufacturer's instructions. In brief, the sample volume was decreased, and the cells were examined with a light microscope and counted with a hemocytometer. Cells were loaded in each channel with a target output of ~ 5000 cells. The cells were partitioned into Gel Beads in Emulsion in the Chromium Controller instrument, where cell lysis and barcoded reverse transcription of RNA were performed. cDNA was generated, amplified, and quality-assessed using an Agilent 4200 system. Individual libraries were pooled for sequencing on the Illumina NovaSeq 6000 platform with 150 bp paired-end reads with assistance from AccuraMed Technology Limited (Guangzhou, China).

For scRNA-seq cell-clustering and annotation, Cellranger (10 × Genomics, v6.1.2) was used for demultiplexing and generation of barcodes and count matrices. Both mouse and human PBMC matrix data were analyzed using Seurat (v.4.1.1) in R4.1.2. The single-cell sequencing data were imported into Rstudio to create seurat object according to the minimum number of cells greater than 3 and the minimum number of genes greater than 200.

For spleen samples, the cells were selected for a number of genes over 1000, less than 4000, a number of total counts less than 25,000 (Spl CpG-B) and 20000 (Spl PBS) and the percentage of mitochondrial genes was less than 10%. The number of starting cells in the Spl PBS sample was 8018, and the number of cells after QC was 7135. The number of starting cells in the Spl CpG-B sample was 7031, and the number of Spl CpGB cells after QC was 6114.

For pLN samples, the cells were selected for a number of genes over 1000, less than 6000, a number of total counts less than 60,000 and a percentage of mitochondrial genes less than 10%. The number of starting cells in the pLN PBS sample was 3167, and the number of cells after QC was 2093. The number of starting cells in the pLN CpG-B sample was 7767, and the number of pLN CpG-B cells was 6963.

For PLF samples, the cells were selected for a number of genes over 1000, less than 6000, a number of total counts less than 60,000 and a percentage of mitochondrial genes less than 10% (PLF PBS) and 5% (PLF CpG-B). The number of starting cells in the PLF PBS sample was 3805, and the number of cells after QC was 3590. The number of starting cells in the PLF CpG-B sample was 5003, and the number of PLF CpG-B cells after QC was 3565.

For pooled samples, the cells were selected for a number of genes over 1000, less than 3000, a number of total counts less than 30,000 (Spl CpG-B), and a percentage of mitochondrial genes was less than 5%. The number of starting cells in the pooled PBS sample was 6408, and the number of cells after QC was 6056. The number of starting cells in the pooled CpG-B sample was 9411, and the number of Spl CpGB cells after QC was 8478.

For human samples, the cells were selected for a number of genes over 1000, less than 4000, a number of total counts less than 30,000 and a percentage of mitochondrial genes less than 10%. The number of starting cells was 4743, and the number of cells after QC was 3689 (PBS). The number of starting cells was 4058, and the number of cells after QC was 3299 (CPG-B).

Furthermore, we removed potential doublets using the DoubletFinder package (version 2.0.2) of R. Data were normalized using the 'LogNormalize' method (scale factor 10,000) of Seurat's Normalize-Data function. The top 2000 highly variable genes (HVGs) from the normalized expression matrix were identified, centered, and scaled before we performed principal component analysis (PCA). The batch effects were removed by the Harmony package (version 1.0) of R. PCA was performed in Seurat, and the top 10 dimensions and resolution set at 0.5 were used for UMAP. The clustering analysis was performed based on the integrated joint embedding produced by Harmony with the Louvain algorithm. For annotation of the cell clusters, differentiating markers for each cluster were identified using the Wilcoxon rank sum test implemented in Seurat's FindAllMarkers function (min.pct = 0.25, logfc.threshold = 0.25). The cell clusters were annotated based on the DEGs and well-known cellular markers from the literature.

For RNA-velocity, analysis of the cellular trajectory by RNA velocity was performed using the package scVelo (v0.2.4) via dynamical modeling. For estimation of the RNA velocities in DCs subtypes, velocyto was used to distinguish unspliced and spliced messenger RNAs in each sample. The Python package scVelo was then used to recover the directed dynamic information by leveraging RNA-splicing information. Specifically, the data were first normalized using the filter_and_normalize function with the following parameter settings: min_shared_counts 30, n_top_genes 2000. The first and second order moments were computed for velocity estimation using the moments function with the following parameter settings: n_pcs 30 and n_neighbors 30. The velocity vectors were obtained using the velocity function of dynamical modeling. Finally, the velocities were visualized in the UMAP embedding via the UMAP function with default parameters.

For regulatory network inference, a single-cell regulatory network for each subcluster was constructed with the SCENIC Python workflow. Specifically, GRNBoost2 (https://github.com/tmoerman/arboreto) in pySCENIC (v0.11.2) was applied to infer gene regulatory networks from the raw count data. Then, potential direct-binding targets (regulons) were selected based on DNA motif analysis. Finally, gene regulatory network activity for individual cells was identified. For the identification of the different regulators for each subcluster in the groups, the regulon activity was averaged. A regulon-group heatmap was generated with the R pheatmap package. Moreover, the specific regulators of each subcluster were the union of the predicted regulator lists and marker gene lists and were used to generate a specific regulator heatmap with the ComplexHeatmap R package (v2.10.0).

## Human samples

Informed consent was obtained from all subjects. There were no exclusion criteria for healthy volunteers. Peripheral blood mononuclear cells (PBMCs) were isolated from whole blood by using Ficoll (Solarbio, #P8900) gradient centrifugation. PBMC DCs were sorted as Lin⁻ (CD3/CD19/CD14) CD123ˡᵒCD11C⁺HLA-DR⁺ from freshly isolated PBMCs (purity > 93%).

For single-cell analysis, PBMCs from four healthy male volunteers were plated individually in complete RPMI-1640 (RPMI-1640 supplemented with 10% FCS, 1% L-glutamate, 1% penicillin/streptomycin, and 55 μM 2-mercaptoethanol), and treated with PBS or 100 nM CpG-B (InvivoGen, #tlrl-2006) for 16 h. Then, DCs (Lin⁻ CD123ˡᵒ CD11C⁺HLA-DR⁺) were subjected to FACS sorting and pooled for scRNA-seq library preparation.

For qPCR analysis, PBMC DCs were stimulated for 14 h with 1 μg/mL Pam3CSK4 (InvivoGen, #tlrl-pms), 100 ng/mL LPS (Sigma-Aldrich, #L2654-1MG), 500 ng/mL Poly(I:C) (InvivoGen, #tlrl-picw), 1 μg/mL R848 (InvivoGen, #tlrl-r848), or 100 nM CpG-B (InvivoGen, #tlrl-2006).

## Statistical analysis

Statistical analysis was performed with Prism 10 (GraphPad) software unless otherwise indicated. Survival rate statistics were analyzed according to the Log-rank Mantel−Cox test. Weight change statistics were analyzed according to two-way ANOVA with Sidak's multiple comparisons test. Comparisons between two groups were performed

by the two-tailed Student's *t* test. Comparisons between more than two groups were performed by one-way ANOVA with Dunnett's multiple comparisons test. *P*- value of 0.05 or less was considered significant. ns, not significant; *, $P < 0.05$; **, $P < 0.01$; ***, $P < 0.001$; ****, $P < 0.0001$ (two-tailed test).

## Study approval

The human study was approved by the Ethics Committee of the First Affiliated Hospital, Sun Yat-sen University, Guangzhou, Guangdong, China (Approval No. 2022-436). Written informed consent was received prior to participation. The animal studies was approved by approved by the Animal Ethics Committee of Sun Yat-sen University, Guangzhou, Guangdong, China.

## Reporting summary

Further information on research design is available in the Nature Portfolio Reporting Summary linked to this article.

## Data availability

The scRNA-seq data and the CUT&TAG data generated in this study have been deposited in the Gene Expression Omnibus database under accession code GSE260857, GSE260858, GSE260859, GSE260860, GSE260861, GSE275286, GSE275287. The data in Fig. 6e (top panel) were previously published[18], and downloaded from the Gene Expression Omnibus accession number (GSE94820). The dataset from GSE94820 was analyzed using the R software package Seurat (V4.0), available from CRAN. Source data are provided in this paper.

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

## Acknowledgements

We thank Dr. Ananda Goldrath for the LM-OVA and VSV. We are grateful to Dr. Xiaojun Xia for B16-OVA and critically reading the manuscript. We thank Dr. Weibin Cai for technical assistance at the animal facilities. This research was supported by the National Key R&D Program of China (2018YFA0508300 to C.Y.Y.), the National Natural Science Foundation of China (31970832 to C.Y.Y.), the Guangdong Innovative and Entrepreneurial Research Team Program (2016ZT06S638 to C.Y.Y.), National Natural Science Foundation of China (T2422018 to A.R.W.) and the Hong Kong Innovation and Technology Commission (ITCPD/17-9 to A.R.W.).

## Author contributions

X.L., L.W., and Y.X. designed and performed the experiments, analyzed and interpreted the data, and wrote the manuscript. H.N., J.H., and K.Y. performed the experiments. W. Z. helped to supervise the CUT&TAG experiments. J.Z. helped to supervise the human experiments and supervised patient protocol logistics. A.R.W. and J.Y. helped to supervise the single-cell RNA-seq experiments. C.Y.Y. designed all the experiments, interpreted the data, wrote the manuscript, contributed to funding, and supervised the entire study.

## Competing interests

The authors declare no competing interests.
