## [Transparent Peer Review file · Nature Communications]

The transcription factor EHF promotes the maturation and immunosuppression of conventional dendritic cells

Corresponding Author: Dr Cliff Yang

Version 0:

Reviewer comments:

Reviewer #1

(Remarks to the Author)

Summary: In this study, the authors investigated the role of EHF, an ETS-family transcription factor, in dendritic cell (DC) function using mouse disease models, CUT&Tag analysis, and single-cell RNA sequencing. Using $Ehf^{\Delta CD11c}$ mice, they found that cDCs developed normally under physiological conditions. However, in a DSS-induced colitis model, these mice exhibited exacerbated disease with significantly increased IL-17A-producing CD4⁺ T cells. In a CpG-induced Th1 model, the number of OT-II cells and their IFN- γ production were also elevated, whereas no differences were observed in IL-4 production in a papain-induced Th2 model. These findings were recapitulated in an in vitro co-culture system of cDC1/cDC2 with OT-II T cells.

Stimulation with CpG and R848 upregulated *Ehf* expression in cDC1 and cDC2, while poly(I:C) suppressed it. scRNA-seq analysis revealed that *Ehf* expression is restricted to CCR7⁺ cDCs. In vitro, GM-CSF and IFN- γ downregulated *Ehf* in cDCs. In vivo, cDC1 and cDC2 from CpG-treated $Ehf^{\Delta CD11c}$ mice showed reduced expression of *Ccr7*, *Cd200*, and *Pd-I1*, while overexpression of *Ehf* enhanced their expression. CUT&Tag experiments suggested that EHF directly binds to and regulates these genes. Human scRNA-seq data further confirmed that EHF is preferentially expressed in CCR7⁺ cDCs, which also express maturation and inhibitory markers.

Collectively, the authors conclude that EHF promotes the maturation of cDC1 and cDC2, while concurrently restraining their capacity to activate Th1 and Th17 cells.

The experiments are generally well designed and the results are mostly convincing. However, additional functional validation would be necessary to firmly support the conclusion that EHF directly regulates its target genes.

Major comments

1. The authors provide a broad overview of dendritic cell (DC) classification and the transcriptional regulators involved in DC development. While this background is informative, some of the content is somewhat redundant and could be streamlined. Additionally, the introduction offers only minimal context regarding EHF and its potential role in DC function. To strengthen the narrative and clarify the rationale of the study, I suggest reorganizing the introduction to focus more concisely on key aspects of DC biology relevant to the study's aim, and expanding on prior knowledge of EHF's role in immune cells, particularly in DCs if available.

2. The authors present CUT&Tag and qPCR validation to suggest that EHF directly regulates genes such as *Ccr7* and *Cd274*. While the data are highly suggestive, further experiments — such as functional assays (e.g., luciferase reporter assays or motif mutagenesis) — would be required to definitively establish direct transcriptional control.”

Specific comments

3. The authors describe the DSS-induced colitis model as an autoimmune disease model. However, DSS-induced colitis is typically considered a chemically induced epithelial injury model that triggers innate immune activation, rather than a classical T cell-mediated autoimmune model. If the authors regard this model as autoimmune, it would be helpful to clarify how T cell-mediated autoimmune responses contribute to disease development in this context, and to cite relevant literature supporting this classification.
4. In the B16-OVA melanoma model, CTL responses were enhanced in $Ehf^{\Delta CD11c}$ mice (Figure 1), suggesting that EHF in DCs plays a critical role in regulating CTL function in vivo. However, in the in vitro co-culture experiments, DCs lacking EHF did not affect the development or activation of CTLs. This apparent discrepancy between in vivo and in vitro findings is somewhat confusing. Could the authors clarify how they interpret this inconsistency? For example, are additional in vivo factors or microenvironmental cues (e.g., cytokines or TME signals) necessary for EHF-mediated modulation of CTL responses?
5. In Figure 2, the data suggest that $Ehf^{\Delta CD11c}$ DCs promote enhanced Th1 and Th17 responses, while Th2 responses appear unaffected. This indicates a selective inhibitory role for EHF in Th1/Th17 development. It would strengthen the manuscript if the authors could discuss potential mechanisms underlying this selectivity. For example, are there differences in how EHF influences the expression of co-stimulatory molecules or cytokines (e.g., IL-12, IL-6, IL-23) that preferentially affect Th1/Th17 polarization but not Th2?
6. The authors screened various cytokines and found that GM-CSF and IFN- γ downregulate Ehf expression in DCs in vitro. However, the rationale for focusing on cytokines as potential negative regulators of Ehf is not clearly explained. Could the authors elaborate on why cytokines were prioritized in this analysis over other potential regulatory factors (e.g., PRRs, transcriptional repressors)? Additionally, are GM-CSF and IFN- γ expressed and functionally involved in the downregulation of Ehf in the CpG injection model in vivo? Addressing these points would help clarify the relevance of the in vitro findings to the in vivo system.
7. In Figure 4A, the number of splenic cDCs appears to be reduced in $Ehf^{\Delta CD11c}$ mice. However, this finding is not mentioned or discussed in the text. Please consider including this result in the main text and providing a possible interpretation. For example, is this reduction due to impaired survival, reduced development, or altered migration of cDCs in the absence of EHF?
8. In Figure 4B, the authors show reduced expression of CCR7, CD200, PD-L1, IRF4, and c-Rel in cDC1 and cDC2 from $Ehf^{\Delta CD11c}$ mice after CpG stimulation. To better understand the regulatory role of EHF, it would be helpful to also present the expression levels of these molecules under basal (unstimulated) conditions. This would clarify whether EHF controls their expression constitutively or primarily in response to CpG stimulation.
9. In Figure 4D, the expression levels of CCR7 and CD200 are assessed in BMDCs, whereas the surrounding experiments primarily utilize the cDC-FL-Notch in vitro system. Could the authors clarify why BMDCs were used in this particular experiment instead of cDC-FL-Notch-derived cells? This inconsistency may raise concerns about comparability, especially given the differences in differentiation and maturation status between BMDCs and FL-derived cDCs.
10. The authors show that EHF expression in human cDCs is enriched in CCR7⁺ populations, which also express maturation and inhibitory markers. However, the functional relevance of these human CCR7⁺EHF⁺ cDCs in shaping Th1, Th2, or Th17 responses remains unclear. Incorporating functional assays — such as T cell co-culture experiments — would help validate whether EHF plays a comparable immunomodulatory role in human cDCs and strengthen the translational relevance of the findings.
11. The term “regulate” in the title is somewhat ambiguous, as it may refer to either positive or negative regulation. I recommend using a more specific verb that clearly reflects the functional role of EHF in cDCs, such as “promotes,” “restrains,” or “modulates,” depending on the authors’ intended emphasis. Additionally, the term “dendritic cells” is broad, while this study focuses specifically on conventional dendritic cells (cDC1 and cDC2). For clarity and precision, I suggest explicitly referencing cDCs in the title.
12. The graphical abstract effectively captures the core message of the study but is somewhat simplified. Enhancing the figure by including key mechanistic elements — such as the role of EHF in regulating CCR7, PD-L1, or its impact on Th1/Th17 responses — would improve its clarity and impact. Please consider refining the graphical abstract to better reflect the depth of the findings.

Reviewer #2

(Remarks to the Author)

The manuscript by Liu et al. aims at characterizing the role of the transcription factor EHT in dendritic cells. EHT is a member of the ETS family, which has been implicated in DC development. They report that EHT is expressed in a subset of DCs, which are characterized by the expression of CCR7, CD200 and FSCN1, or mRegDCs. Deletion of EHT in DCs led to worsening of colitis, and better control of B16 tumour growth and Listeria infection. They mainly attributed these phenotypes to changes in CD4 T cells, although changes are often mild.

Understanding of CCR7+ CD200+ DCs are regulated is important, given their emerging central role in controlling T-cell responses. The data generated is interesting in places, but deleting EHF in DCs has only a subtle impact on DC phenotype and subsequent T-cell responses, reducing the enthusiasm for this manuscript. In addition, while experiments are well designed, the conclusions are not always fully supported by the data.

Some specific comments that might help improve the manuscript:

- 1- In Figure 1, the authors described an increase in CD8 T cells (CTLs or OTI) following B16-OVA in Ehf depletion in DCs. But in figure 2, they do not find any difference in CD8 T cell expansion/activation when activated by WT or EhfKO DCs. How do the authors explain this?
- 2- The acquisition of the maturation programme from DC1/DC2 to CCR7+ CD200+ DCs is dependent on the uptake of tumour antigens (for example, Maier et al. Nature 2020), as the authors also suggest by incubating DCs with dead cells in Figure 3. How does EHF expression/function fit into this? Is there a synergy between antigen uptake and TLR signalling for EHF expression? Is it the actual uptake that's important? What happens if antigen uptake is inhibited?
- 3- In multiple parts of the manuscript, but I will take Fig.S1 as an example, I disagree with the way protein expression is quantified from flow cytometry data. MFI should be reported if there is just one peak; mean can be used if the distribution is normal, but this is often not the case in flow data, so median (or g-mean) should rather be used. The authors seem to have reported the mean throughout the manuscript. In fig.S1K or S1N, there are clear positive and negative populations. Therefore, quantifying the percentage of positive cells is more relevant, or the MFI of the positive cells only. In addition, the difference in MFI presented, albeit statistically significant, is often very small. What is the functional/biological relevance of expressing slightly more CD44 per cell, for example? There is not direct link between slight differences in CD44 expression in a given cell and its function. Finally, in Fig.S1N, the CD44 negative fraction has a higher MFI in the KO than WT condition, making comparing MFI problematic. Why is that the case? Was the same number of cells stained?
- 4- In Fig.S2G, the percentage of IL17+ and IL22+ CD4 T cells is very small. Where the cells restimulated in vitro? What is their percentage in the colon?
- 5- In Figure 3, the authors demonstrate that EHF expression is induced by TLR9 activation and dependent in Myd88. Is there a difference in mRegDC generation in Myd88KO mice at steady state and in disease (or at least after CpG treatment)?
- 6- In Figure 3C, in the CpG treated condition, it is unclear which cluster corresponds to which cell type, especially in the myeloid area. For example, where are the DC1?
- 7- In Figure 3G, what is the effect of IL-17 on Ehf expression? As the authors show that EHF inhibits the expression of Th1 and Th17 responses and IFN γ decreases EHF expression, I wonder whether that's also the case for IL-17. It seems to be that EHF is part of a negative feedback loop, rather than controlling mRegDC maturation per se.
- 8- Similarly as point 3, in Figure 4B, the differences in CCR7 expression, especially in DSS treated conditions, is subtle and it is unclear if that's biologically relevant. In Fig.S4A, functional migration assay also shows only mild differences and the migratory defect is also present without cytokines, it is therefore unclear how much of the difference in migration is due to differences in CCR7 expression. It will be helpful to use different doses of CCL19/21, and characterizing the migration to another chemokine for which the receptor is not affected by EHF deletion.
- 9- In Figure 5, what are the genes driving the GO enrichment? A table would be useful.
- 10- In Fig.6, only peritoneal cavity is tested as a non-lymphoid organ, I would be careful in generalising. Quantifying the expression of EHF within DC subsets in the colon or tumor in disease settings as in Fig.1 would help with this. Similarly, quantifying the frequency/activation potential of mRegDCs vs other DCs in the same disease models in mice with DC-specific depletion of EHF would strengthen the role of EHF in mRegDCs.
- 11- EHF is expressed in different CCR7+ clusters (cluster 2 vs cluster 3) in the spleen vs LN (Fig6D). Do the authors know why this is the case? Does it correlate with differences in CCR7/CD200/CD274 expression? The authors present the overall integrated data in Sup figS6, but given the difference in EHF expression in LN vs spleen, it is important to look at those other markers per site too. This dataset could also be helpful in confirming other points. For example, do the authors see a correlation between TLR or IFN pathways and EHF expression within specific DC subsets?
- 11- In Fig.6, stats on vlnplots are done based on n being a cell. Since the authors have 6 samples, it should rather be done on a sample basis, to be sure that the phenotype is not driven by one sample.
- 12- It should be noted that, while the authors focus on the potential inhibitory role of mReg DCs, whether they have a positive or negative effect on disease outcomes remains unclear, and enrichment of a CCR7+ DC signature is associated with improved survival in multiple cancers (Lee et al, Nat Comms, 2024). It would be interesting for the authors to discuss how their data integrate with this.
- 13- In some places, the number of independent experiments performed is not stated.

Reviewer #3

(Remarks to the Author)

The study by Liu et al. identifies ETS homologous factor (EHF) as a key transcriptional regulator controlling maturation and immunosuppressive functions of conventional dendritic cells (cDCs) following TLR7 and TLR9 activation. Using conditional knockout mice, the authors demonstrate that EHF deficiency in DCs enhances resistance against DSS colitis, bacterial infections, and melanoma growth, primarily by promoting stronger Th1- and Th17-biased CD4+ T-cell responses. Mechanistically, the authors claim that EHF directly regulates the transcription of genes involved in dendritic cell maturation and immunosuppression, including *Ccr7*, *Cd200*, *Cd274* (PD-L1), *Irf4*, and *Rel*, thus highlighting its pivotal role in suppressing inflammatory responses.

This study represents the first comprehensive characterization of EHF's in vivo role in DC biology, expanding our understanding beyond its previously described functions in epithelial and tumor cell biology. The authors performed an extensive amount of experimental work, including the generation and analysis of new conditional knockout mouse models across several in vivo disease settings. Additionally, single-cell RNA sequencing analyses further reinforced their findings,

providing high-resolution insights into EHF expression patterns across DC subsets in mice and humans at steady state and stimulatory conditions. A notable technical advance is the development of a novel monoclonal antibody recognizing EHF, which enabled CUT&TAG analyses to precisely identify putative EHF DNA binding sites.

While this manuscript provides novel insights into the role of EHF in DC biology and is supported by a big and diverse set of experiments, certain aspects of the study feel inconsistent, with some datasets conflicting or not fully aligning with the authors' claims. Some conclusions are stated too strongly relative to the presented data and would benefit from either additional experimental validation or a more nuanced interpretation.

Major points:

Figure 1:

My main concern is the limited analysis of DC2 subsets (cDC2a, cDC2b and DC3): although the data indicate that resident cDC2s express EHF at steady state, this EHF-expressing population was not examined or characterized in detail. Furthermore, DC2 subsets were not analyzed in sufficient depth in the in vivo models, either by flow cytometry or single-cell RNA sequencing. A more detailed investigation of DC2 heterogeneity in WT versus KO mice, both at steady state and in disease models, would greatly strengthen the manuscript's conclusions.

Figure 1L: The authors challenge Ehf CD11c-Cre mice with B16-OVA and show that Ehf-deficient mice have increased survival compared to WT controls. They also demonstrate that, upon OT-I transfer, OT-I cells exhibit decreased PD-1 expression in Ehf-deficient mice, but they do not provide an explanation for this observation. Normally, PD-1 can act as both an activation and exhaustion marker, yet it remains unclear why OT-I cells show reduced PD-1 expression in the absence of EHF and what mechanisms drive this phenotype. Furthermore, the authors should include data on the total numbers of engrafted OT-I cells and perform tetramer staining to assess antigen-specific OT-I populations in WT versus Ehf-deficient mice.

Figure S1J: How did the authors evaluate the expression of IFN α by pDCs using intracellular staining? The use of intracellular antibodies against IFN α is not described in the Materials and Methods section. This information is important, as intracellular staining for IFN α is technically challenging and has only been described by a few groups.

Figure S1O and S1Q: The authors need to include non-treated/non-infected mice as control groups.

Figure 2:

Figure 2A-F: The study would benefit from an analysis that goes beyond cytokine production of OT-II engrafted cells by also assessing their true T helper cell identity through intracellular staining of GATA3, T-bet, Foxp3, and ROR γ t. Furthermore, non-infected/non-treated mice should also be included as controls in this figure in the revised manuscript.

Figure S2F-G: The authors show that Ehf-deficient mice have reduced inflammation when challenged with *C. rodentium*. This finding is unexpected and contrasts with the data shown in Figure 1/S1, where DSS-treated Ehf-deficient mice exhibited stronger inflammation compared to WT mice. Is the phenotype different if the authors analyze *C. rodentium*-infected mice at an earlier time point post-infection (e.g., day 6)? Could day 11 post-*C. rodentium* infection already be too late to observe the increased pro-inflammatory responses that might be expected in Ehf-deficient mice?

Figure S2E: I would recommend moving Figure S2E into Figure S1, as it directly relates to the DSS experiment described in Figure 1 and would be better placed there for clarity and consistency.

Figure 3:

Figure 3A: The authors performed qPCR on cDC1 and cDC2 stimulated in vitro with distinct TLR agonists to measure Ehf expression. They present the data as relative expression compared to the control sample. However, a more appropriate approach would be to normalize Ehf expression to a housekeeping gene. This would not only allow for comparison between the different stimulations but also provide an indication of the Ehf expression levels between cDC1 and cDC2.

Figure 3B: The authors performed scRNA sequencing on splenocytes derived from B6 mice that were either treated with CpG-B or left untreated. In the CpG-B-treated panel, the cDC1 population does not appear to be annotated. Is this because cDC1 cells were absent in CpG-B-treated mice, or is this an annotation error?

Figure 4:

Figure 4A: The authors demonstrate that cDCs derived from Ehf-deficient mice are reduced upon stimulation with CpG-B. It is important to elucidate which specific cDC subset is affected by the loss of EHF in this context. In particular, the authors should assess whether there is a decrease in cDC1, cDC2a, cDC2b, and/or DC3 populations.

Figure 4B-D: The authors demonstrate that cDC1 and cDC2 derived from Ehf-deficient mice show reduced expression of activation markers such as CCR7, CD200, and PD-L1, along with an increase in the transcription factor IRF4 and a decrease in c-Rel upon CpG-B stimulation. While these results are intriguing, they appear somewhat ambiguous, and it is unclear how the authors arrived at these findings. Performing bulk RNA sequencing on cDC1 and cDC2 from WT and Ehf-deficient mice treated with either CpG-B or PBS would help clarify these results and better capture the transcriptional changes associated with the absence of EHF.

Figure 5:

Figure 5A-F: The authors performed a Cut&TAG experiments targeting EHF on BM DC cultures and used a broad

population of cDCs, that contains both, cDC1 and cDC2 subsets to target EHF. Eventhough it gives already a good preliminary idea about the DNA binding regions of EHF, it still overlaps the peaks present in cDC1 and cDC2. A clearer and better approach would be to target ex vivo isolated cDC1 and cDC2 from the spleen that were stimulated with CpG-B or PBS. This would reveal the EHF binding sites in both, steady state as well as activation state and give a more and proufnd idea about how EHF regulates the transcriptional landscape within each cDC subset.

Figure 5E: The authors should tone down their statement (“...we found that EHF may directly promote the transcription of inhibitory molecules such as CCR7, CD200, and PD-L1.”, Line 491–492) regarding EHF specifically regulating the expression of these genes. Most of the depicted EHF DNA-binding sites are located within introns or in upstream/downstream enhancer regions, with only Cd274 showing a peak ~3 kb downstream of the promoter region. Since none of these enhancer regions were functionally evaluated, it cannot be conclusively claimed that EHF directly regulates these target genes.

Version 1:

Reviewer comments:

Reviewer #1

(Remarks to the Author)

The authors have adequately addressed all reviewer concerns and questions, and the revision is acceptable.

As the mediator for Reviewer #3

Overall, the authors' responses to the reviewers' comments are satisfactory, and the major scientific concerns have been adequately addressed. I only note two minor issues where statements in the rebuttal do not appear to be reflected in the revised manuscript text, as detailed below.

1.The authors state in their response that an explanation linking reduced PD-L1 expression on EHF-deficient DCs to lower PD-1 expression on OT-I cells has been added to the manuscript. However, I was unable to identify an explicit statement to this effect in the revised text. Please indicate where this explanation is described or revise the text to clearly state this interpretation.

2.The authors state in their response that details of IFN- α staining have been added to the Methods; however, I could not locate any description of IFN- α staining in the revised manuscript or Supplementary Information. Please clarify where this information is provided, or revise the response accordingly.

Reviewer #2

(Remarks to the Author)

The authors have responded to prior critiques. I do not have any further comments.

We would like to thank all reviewers for their constructive criticism of our manuscript. They have raised important concerns and we have addressed them in the revised manuscript with new experiments. The revisions are described below and also highlighted in the manuscript.

Reviewer #1 (Remarks to the Author):

Summary: In this study, the authors investigated the role of EHF, an ETS-family transcription factor, in dendritic cell (DC) function using mouse disease models, CUT&Tag analysis, and single-cell RNA sequencing. Using *Ehf*^{ΔCD11c} mice, they found that cDCs developed normally under physiological conditions. However, in a DSS-induced colitis model, these mice exhibited exacerbated disease with significantly increased IL-17A-producing CD4⁺ T cells. In a CpG-induced Th1 model, the number of OT-II cells and their IFN-γ production were also elevated, whereas no differences were observed in IL-4 production in a papain-induced Th2 model. These findings were recapitulated in an in vitro co-culture system of cDC1/cDC2 with OT-II T cells.

Stimulation with CpG and R848 upregulated *Ehf* expression in cDC1 and cDC2, while poly(I:C) suppressed it. scRNA-seq analysis revealed that *Ehf* expression is restricted to CCR7⁺ cDCs. In vitro, GM-CSF and IFN-γ downregulated *Ehf* in cDCs. In vivo, cDC1 and cDC2 from CpG-treated *Ehf*^{ΔCD11c} mice showed reduced expression of *Ccr7*, *Cd200*, and *Pd-I1*, while overexpression of *Ehf* enhanced their expression. CUT&Tag experiments suggested that EHF directly binds to and regulates these genes. Human scRNA-seq data further confirmed that EHF is preferentially expressed in CCR7⁺ cDCs, which also express maturation and inhibitory markers.

Collectively, the authors conclude that EHF promotes the maturation of cDC1 and cDC2, while concurrently restraining their capacity to activate Th1 and Th17 cells.

The experiments are generally well designed and the results are mostly convincing. However, additional functional validation would be necessary to firmly support the conclusion that EHF directly regulates its target genes.

Major comments

1. The authors provide a broad overview of dendritic cell (DC) classification and the transcriptional regulators involved in DC development. While this background is informative, some of the content is somewhat redundant and could be streamlined. Additionally, the introduction offers only minimal context regarding EHF and its potential role in DC function. To strengthen the narrative and clarify the rationale of the study, I suggest reorganizing the introduction to focus more concisely on key aspects of DC biology relevant to the study's aim, and expanding on prior knowledge of EHF's role in immune cells, particularly in DCs if available.

Thank you very much for the suggestion. We streamlined the first three paragraphs

into just one. We focused more on mregDCs, ETS family's role in DCs and expanded more on EHF. Unfortunately, there isn't much prior knowledge of EHF's role in DCs except for the human MoDC studies from the same group.

2. The authors present CUT&Tag and qPCR validation to suggest that EHF directly regulates genes such as *Ccr7* and *Cd274*. While the data are highly suggestive, further experiments — such as functional assays (e.g., luciferase reporter assays or motif mutagenesis) — would be required to definitively establish direct transcriptional control.”.

Thank you for this suggestion. We performed the motif mutagenesis in luciferase reporter assays on all putative ETS-binding sites. The results were confirmatory. These data were added in **Fig. 5g** (shown below). We hope this functional assay support our claim of direct transcriptional control.

Specific comments

3. The authors describe the DSS-induced colitis model as an autoimmune disease model. However, DSS-induced colitis is typically considered a chemically induced epithelial injury model that triggers innate immune activation, rather than a classical T cell-mediated autoimmune model. If the authors regard this model as autoimmune, it would be helpful to clarify how T cell-mediated autoimmune responses contribute to disease development in this context, and to cite relevant literature supporting this classification.

We agree that the DSS-induced colitis model is initiated by innate immunity, rather than T cells. While DSS induces acute colitis in a T-cell independent (RAG knockout) manner, many studies in DSS-induced colitis had shown a steady upregulation of Th1/Th2/Th17 cytokines, aggravation of the inflammatory response by T-cells in wild-type mice with intact innate and adaptive immunity, and Treg's involvement. T-cell related therapies such as anti-TL1A and $\alpha 4\beta 7$, IL-23R antibodies had shown good efficacy in DSS-colitis models and associated effects on T cells. Thus, we still regard this model as autoimmune but not necessarily entirely T-cell-driven. We have refined statements and citations (Ref: 43-47) reflective of this distinction in the text (highlighted). We regret that we cannot cite more literature due to citation limit.

4. In the B16-OVA melanoma model, CTL responses were enhanced in *Ehf*^{ΔCD11c} mice (Figure 1), suggesting that EHF in DCs plays a critical role in regulating CTL function *in vivo*. However, in the *in vitro* co-culture experiments, DCs lacking EHF did not affect the development or activation of CTLs. This apparent discrepancy between *in vivo* and *in vitro* findings is somewhat confusing. Could the authors clarify how they interpret this inconsistency? For example, are additional *in vivo* factors or microenvironmental cues (e.g., cytokines or TME signals) necessary for EHF-mediated modulation of CTL responses?

Thank you for pointing it out. We think there might be two reasons. First, we found that EHF in DCs regulates Th1 response (Fig. 2c, shown below), which directly controls CTL response. However, this CTL-Th1 interplay was difficult to capture in an CTL assay *in vitro* with just CD8⁺ T cells.

Second, we found that EHF is downregulated by IFN γ (new Fig. 3h and 3i, shown below). It is well known that the IFN γ production by CTLs dwarfs that of Th1s. The microenvironmental IFN γ concentration may accumulate more in a cell culture well than what happens *in vivo*. Thus, EHF in DCs may downregulate more quickly in a well with CTLs compared to a well with Th1s. If there is little or no EHF upregulation, there might be not any phenotypical difference between EHF-wildtype and -knockout. This IFN γ -driven negative regulation could also explain the very limited CTL phenotype in infection models, in which IFN γ is massively produced compared to “not-so-hot” tumor model.

Since the in vitro and in vivo results are inconsistent, we err on the side of caution in concluding that EHF is dispensable for CTL response. We have included the explanation for this discrepancy in the discussion.

5. In Figure 2, the data suggest that $Ehf^{\Delta CD11c}$ DCs promote enhanced Th1 and Th17 responses, while Th2 responses appear unaffected. This indicates a selective inhibitory role for EHF in Th1/Th17 development. It would strengthen the manuscript if the authors could discuss potential mechanisms underlying this selectivity. For example, are there differences in how EHF influences the expression of co-stimulatory molecules or cytokines (e.g., IL-12, IL-6, IL-23) that preferentially affect Th1/Th17 polarization but not Th2?

Thank you for this key observation. We had examined cytokine production by FACS and RNA-seq. However, we found no specific pattern for Th1/Th2 related molecules (**Supplementary Fig. 1j**, shown below). EHF did not seem to regulate cytokine production.

We think that a possible mechanism could be the sustained up-regulation of EHF expression in type I and type III immune response. We found that TLR7/8/9 activation via apoptotic cells is a main trigger of EHF upregulation. TLR7/8/9 signaling are well-known to induce Th1/Th17 response, and the increased apoptosis in type I and type III immune response to intracellular pathogens (compared to Type II humoral response to extracellular pathogens) may reinforce EHF expression. Therefore, the differential EHF regulation of Th1/Th17 response could be due to a longer activation of TLR7/8/9 and subsequent sustained EHF expression in DCs. This is now included in the discussion.

6. The authors screened various cytokines and found that GM-CSF and IFN- γ downregulate Ehf expression in DCs in vitro. However, the rationale for focusing on cytokines as potential negative regulators of Ehf is not clearly explained. Could the authors elaborate on why cytokines were prioritized in this analysis over other potential regulatory factors (e.g., PRRs, transcriptional repressors)?

Thank you for these questions. First, we have examined PRRs first (**Fig. 3a**, shown below).

Next, we looked at MyD88 as a transcriptional regulator (**Fig. 3f**, shown below).

Then at last, we turned to cytokines (new **Fig. 3h**, see below).

Additionally, are GM-CSF and IFN- γ expressed and functionally involved in the downregulation of Ehf in the CpG injection model in vivo? Addressing these points would help clarify the relevance of the in vitro findings to the in vivo system.

In the original **Fig. 3i**, the anti-GM-CSF antibodies blocking experiments were actually performed in vivo. In this revision, we added the anti-IFN γ antibodies blocking in vivo experiments (new **Figure 3i**, shown below). Both in vivo and in vitro findings matched.

7. In Figure 4A, the number of splenic cDCs appears to be reduced in Ehf $^{\Delta CD11c}$ mice. However, this finding is not mentioned or discussed in the text. Please consider including this result in the main text and providing a possible interpretation. For example, is this reduction due to impaired survival, reduced development, or altered migration of cDCs in the absence of EHF?

Thank you for pointing this out. Although we mention this phenotype in the original text, we did a poor job explaining it. Since DC development and cell numbers were normal in EHF-deficient mice (**Supplementary Fig. 1d**, shown below), we reasoned it could be altered migration.

Thus, we performed a DC trafficking experiment to confirm that (new **Supplementary Fig. 4a**, see below).

This was also consistent with a reduced CCR7 expression by EHF-deficient DCs (new Fig. 4d, see below).

We have now revised the description accordingly in the text (highlighted).

8. In Figure 4B, the authors show reduced expression of CCR7, CD200, PD-L1, IRF4, and c-Rel in cDC1 and cDC2 from *Ehf Δ CD11c* mice after CpG stimulation. To better understand the regulatory role of EHF, it would be helpful to also present the expression levels of these molecules under basal (unstimulated) conditions. This would clarify whether EHF controls their expression constitutively or primarily in response to CpG stimulation.

We have now added the expression levels of these molecules under basal condition in the new **Fig. 4d** and **4f** (shown below). They showed that EHF's role was not constitutive and primarily in response to TLR7/8/9 stimulation.

9. In Figure 4D, the expression levels of CCR7 and CD200 are assessed in BMDCs, whereas the surrounding experiments primarily utilize the cDC-FL-Notch in vitro system. Could the authors clarify why BMDCs were used in this particular experiment instead of cDC-FL-Notch-derived cells? This inconsistency may raise concerns about comparability, especially given the differences in differentiation and maturation status between BMDCs and FL-derived cDCs.

We have performed the overexpression in both BMDCs and cDC^{FL-Notch}. The results are similar (see below). We chose the data from BMDCs because cDC^{FL-Notch} did not seem to express much CD200 and PD-L1 at the first place. To be consistent, we have now replaced the BMDC data in the new Fig. 4d with data from cDC^{FL-Notch}.

10. The authors show that EHF expression in human cDCs is enriched in CCR7⁺ populations, which also express maturation and inhibitory markers. However, the functional relevance of these human CCR7⁺EHF⁺ cDCs in shaping Th1, Th2, or Th17 responses remains unclear. Incorporating functional assays — such as T cell co-culture experiments — would help validate whether EHF plays a comparable immunomodulatory role in human cDCs and strengthen the translational relevance of the findings.

Yes, the human T cell experiment would strengthen the translational relevance of the findings. We collected more human samples and looked at the CCR7 expression in PMBCs (see below).

Consistent with our previous observation, there was only ~6% of CCR7⁺ DCs (gated from all DCs). We are only allowed by our ethnic committee to extract a maximum of

100cc of blood from each volunteer. Thus, in theory that is about 40,000 cells per sample (before sorting). However, after MACS purification and FACS sorting, we only collected a few thousand CCR7⁺ DCs. These CCR7⁺ DC were not so viable and they failed to activate T cells.

We simply do not have enough DCs cells to perform the Th1/2/17 experiments, given the limitation on blood volume. Given human T cells are allogenic, we cannot combine individual samples. We have also tried different titration of CCR7 antibodies, and the gating looked similar.

Perhaps PBMCs are not like murine spleens, they are always CCR7⁺ or cannot get activated as much. In light of this, we have taken care in the text to avoid any claims of translational relevance.

11. The term “regulate” in the title is somewhat ambiguous, as it may refer to either positive or negative regulation. I recommend using a more specific verb that clearly reflects the functional role of EHF in cDCs, such as “promotes,” “restrains,” or “modulates,” depending on the authors’ intended emphasis. Additionally, the term “dendritic cells” is broad, while this study focuses specifically on conventional dendritic cells (cDC1 and cDC2). For clarity and precision, I suggest explicitly referencing cDCs in the title.

Thank you for the suggestion. We have changed the title to “The transcription factor EHF **promotes** the maturation and immunosuppression of **conventional** dendritic cells”.

12. The graphical abstract effectively captures the core message of the study but is somewhat simplified. Enhancing the figure by including key mechanistic elements — such as the role of EHF in regulating CCR7, PD-L1, or its impact on Th1/Th17 responses — would improve its clarity and impact. Please consider refining the graphical abstract to better reflect the depth of the findings.

We have refined the graphical abstract (see below) with more details accordingly.

Reviewer #2 (Remarks to the Author):

The manuscript by Liu et al. aims at characterizing the role of the transcription factor EHT in dendritic cells. EHT is a member of the ETS family, which has been implicated in DC development. They report that EHT is expressed in a subset of DCs, which are characterized by the expression of CCR7, CD200 and FSCN1, or mRegDCs. Deletion of EHT in DCs led to worsening of colitis, and better control of B16 tumour growth and Listeria infection. They mainly attributed these phenotypes to changes in CD4 T cells, although changes are often mild.

Understanding of CCR7+ CD200+ DCs are regulated is important, given their emerging central role in controlling T-cell responses. The data generated is interesting in places, but deleting EHF in DCs has only a subtle impact on DC phenotype and subsequent T-cell responses, reducing the enthusiasm for this manuscript. In addition, while experiments are well designed, the conclusions are not always fully supported by the data.

Some specific comments that might help improve the manuscript:

1- In Figure 1, the authors described an increase in CD8 T cells (CTLs or OTI) following B16-OVA in Ehf depletion in DCs. But in figure 2, they do not find any difference in CD8 T cell expansion/activation when activated by WT or EhfKO DCs. How do the authors explain this?

Thank you for pointing it out. We think there might be two reasons. First, we found that EHF in DCs regulates Th1 response (**Fig. 2c**, shown below), which directly controls CTL response. However, this CTL-Th1 interplay was difficult to capture in an CTL assay *in vitro* with just CD8⁺ T cells.

Second, we found that EHF is downregulated by IFN γ (new **Fig. 3h** and **3i**, shown below). It is well known that the IFN γ production by CTLs dwarfs that of Th1s. The microenvironmental IFN γ concentration may accumulate more in a cell culture well than what happens in vivo. Thus, EHF in DCs may downregulate more quickly in a well with CTLs compared to a well with Th1s. If there is little or no EHF upregulation, there might be not any phenotypical difference between EHF-wildtype and -knockout. This IFN γ -driven negative regulation could also explain the very limited CTL phenotype in infection models, in which IFN γ is massively produced compared to “not-so-hot” tumor model.

Since the in vitro and in vivo results are inconsistent, we err on the side of caution in concluding that EHF is dispensable for CTL response. We have included the explanation for this discrepancy in the discussion.

2- The acquisition of the maturation programme from DC1/DC2 to CCR7+ CD200+ DCs is dependent on the uptake of tumour antigens (for example, Maier et al. Nature 2020), as the authors also suggest by incubating DCs with dead cells in Figure 3. How does EHT expression/function fit into this? Is there a synergy between antigen uptake and TLR signalling for EHF expression? Is it the actual uptake that's important? What happens if antigen uptake is inhibited?

Thank you for suggesting this. We performed an antigen uptake inhibition experiment (new **Fig. 3e**, see below). EHF expression was regulated by antigen uptake.

3- In multiple parts of the manuscript, but I will take Fig.S1 as an example, I disagree with the way protein expression is quantified from flow cytometry data. MFI should be

reported if there is just one peak; mean can be used if the distribution is normal, but this is often not the case in flow data, so median (or g-mean) should rather be used. The authors seem to have reported the mean throughout the manuscript.

We are sorry for the misunderstanding. We always used genomic mean for all analysis throughout the manuscript. It was stated in our original methods: “The MFI was calculated according to the genomic mean by FlowJo software.”

In fig.S1K or S1N, there are clear positive and negative populations. Therefore, quantifying the percentage of positive cells is more relevant, or the MFI of the positive cells only. In addition, the difference in MFI presented, albeit statistically significant, is often very small. What is the functional/biological relevance of expressing slightly more CD44 per cell, for example? There is not direct link between slight differences in CD44 expression in a given cell and its function. Finally, in Fig.S1N, the CD44 negative fraction has a higher MFI in the KO than WT condition, making comparing MFI problematic. Why is that the case? Was the same number of cells stained?

Thank you for pointing it out. We have reanalyzed the data based on positive and negative gating, and the findings were similar (new **Supplementary Fig. 1m, n, p**).

4- In Fig.S2G, the percentage of IL17⁺ and IL22⁺ CD4 T cells is very small. Where the cells restimulated in vitro? What is their percentage in the colon?

Yes, they were restimulated in vitro. We have now added the percentages in the colon (new **Supplementary Fig. 2h**, see below). The percentages are increased 10-fold in the colon compared to the mLN. Thank you for directing us to look at the colon!

5- In Figure 3, the authors demonstrate that EHF expression is induced by TLR9 activation and dependent in Myd88. Is there a difference in mRegDC generation in Myd88KO mice at steady state and in disease (or at least after CpG treatment)?

There was no difference in CCR7⁺ DCs in MyD88-deficient mice during steady state. However, after CpG-B treatment, there was a significant difference in CCR7, CD200 and PD-L1 expression (new **Supplementary Fig. 3c-d**).

6- In Figure 3C, in the CpG treated condition, it is unclear which cluster corresponds to which cell type, especially in the myeloid area. For example, where are the DC1?

Thank you for noticing this important point. After CpG-B treatment, scRNA-seq cannot differentiate CCR7⁺ DC1 and DC2, as all mature cDCs become just one cluster of CCR7⁺ DCs. Cluster analysis suggested that transcriptional programming of CCR7⁺ DC1 and DC2 are very similar. In FACS, it is still possible to differentiate them using traditional surface markers. However, there is a marked decrease in XCR1 expression by cDC1s (Supplementary Information, Gating strategy, see below).

The sc-RNA-seq and FACS profiling suggests that the blurring of DC1/DC2 programming after CpG-B activation and EHF upregulation. This population of XCR1^{int} DCs may be of similar origin to the previously reported regulatory CD103^{int} DCs found in neonatal mice (*Sci. Immunol.* 2023).

This is also consistent with our finding that EHF repressed DC-lineage transcriptional factor such as IRF4, thus blurring the boundary between DC1 and DC2 when EHF is expressed (**Fig. 4f**, see below). We have now added related discussion in the text.

7- In Figure 3G, what is the effect of IL-17 on Ehf expression? As the authors show that EHF inhibits the expression of Th1 and Th17 responses and IFN γ decreases EHF expression, I wonder whether that's also the case for IL-17. It seems to be that EHF is part of a negative feedback loop, rather than controlling mRegDC maturation per se.

We have now added the IL-17 data. However, it had no effect (**Fig. 3h**, see below).

We found no specific pattern for Th1/Th17 related molecules on EHF regulation. EHF seems to be broadly immunosuppressive in terms of its downstream targets. Thus, we think that a possible mechanism for specific Th1/Th17 inhibition could be the sustained up-regulation of EHF expression in type I and type III immune response. We found that TLR7/8/9 activation via apoptotic cells is a main trigger of EHF upregulation. TLR7/8/9 signaling are well-known to induce Th1/Th17 response, and the increased apoptosis in type I and type III immune response may reinforce EHF expression. Therefore, the differential EHF regulation of Th1/Th17 response could be due to a longer activation of TLR7/8/9 and subsequent sustained EHF expression in DCs. This is now included in the discussion.

8- Similarly as point 3, in Figure 4B, the differences in CCR7 expression, especially in DSS treated conditions, is subtle and it is unclear if that's biologically relevant.

We agree that CCR7 defect is subtle if we just looked at MFI. However, if we look at the FACS plots with CCR7/CD200 gating, it might be more obvious (new **Fig. 4b**, see below).

In Fig.S4A, functional migration assay also shows only mild differences and the migratory defect is also present without cytokines, it is therefore unclear how much of the difference in migration is due to differences in CCR7 expression. It will be helpful to use different doses of CCL19/21, and characterizing the migration to another chemokine for which the receptor is not affected by EHF deletion.

We redid the transwell assay with a concentration gradient of CCL19/21 (**Supplementary Fig. 4a**). The effect was dose-dependent. We also had CCL2 as a control, and it is not affected by EHF deletion.

9- In Figure 5, what are the genes driving the GO enrichment? A table would be useful.

We have added the tables in **Supplementary Fig. 5e** (see below) for your information.

Description	Count	Gene ID
Leukocyte migration	43	Tnfsf4/F11r/Slamf9/Abr/Ccl2/Ccl5/Ccl3/Ccl4/Ccr7/Pecam1/Cd300a/Edn1/Lrch1/Ext1/Rac2/Pdgfb/Cd47/Dusp1/Tnf/Vav1/Cd74/Gnt1/Ptprj/Il1b/Sirpa/Mmp9/Lyn/Pde4b/Cdc42/Kit/Hrh1/Rps19/Rhog/Mapk3/Ilgam/Myo9b/Smpd3/St3gal4/Cxcr5/Cadm1/Igga9/Cx3cr1/Ccr1
Mononuclear differentiation	45	Ptprc/Mr1/Tnfsf4/Ly9/Slamf9/Itpkb/Tnfaip3/Lilrb4a/Zbtb7a/Polm/Il12b/Csf2/Fnlp1/Rara/Ccr7/Stat3/Batf/Cd83/Il3ra/St3gal1/Nfam1/Socs1/Plcl2/Vav1/Sos1/Egr1/Cd74/Malt1/Ptprj/Cd44/Il1b/Tpd52/Lyn/Tox/Chd7/Zc3h12a/Runx3/Rbpj/Kit/Lfng/Foxp1/Relb/Bcl3/Pou2f2/Il4i1
Bacterial response	39	Ly96/Slc11a1/Ncf2/Tnfsf4/Fcgr4/Tnfaip3/Lilrb4a/Sbno2/Cactin/Mir146/Il12b/Tnfp1/Igtp/Cd68/Abr/Ccl2/Ccl5/Mir142/Rara/Ccr7/Nfkbia/Mtdh/Cyrb/Tnf/Malt1/Il1b/Sirpa/Lyn/Pde4b/Zc3h12a/Cd36/Tlr1/Tlr6/Scarb1/Foxp1/Irak2/Zfp36/Mapk3/Cx3cr1
Immune activation	34	Slc11a1/Ptprc/Tnfsf4/Cd244a/Ly9/Tnfaip3/Lilrb4a/icosl/Sbno2/Il12b/Abr/Supt6/Rara/Ccr7/Stat3/Cd300a/Batf/Lcp1/St3gal1/Rac2/Plcl2/Cd74/Malt1/Lyn/Zc3h12a/Clnk/Kit/Lfng/Foxp1/Relb/Bcl3/Ilgam/Trex1/Cx3cr1
Immune inhibition	40	Ptprc/Tnfsf4/Tnfaip3/Lilrb4a/Adora2a/Cactin/Il12b/Cd68/Abr/Ccl2/Rara/Cd300a/Lrch1/Tarbp2/Socs1/Cd47/Cblb/Samsn1/Dusp1/Tnf/Plcl2/Cd74/Ptprj/Dgkz/Cd44/Lyn/Zc3h12a/Smpd3/Runx3/Clnk/Oas3/Rps19/Il4i1/Htra1/Ubash3b/Nlr1/Tcta/Trex1/Cx3cr1/Ccr1
Inflammatory response	34	Tnfsf4/Esr1/Tnfaip3/Cdk19/Adora2a/Sbno2/Il12b/Tnfp1/Abr/Ccl5/Ccl3/Nfe2l1/Ccr7/Nfkbia/Pbk/Mefv/Cd47/Tnf/Fem1a/C3/Pik3ap1/Cd44/Il1b/Spata2/Lyn/Smpd3b/Tlr6/Foxp1/Gpr4/Rps19/Zfp36/Nlr1/Trex1/Cx3cr1
T-cell priming	32	Ptprc/Tnfsf4/Cd244a/Itpkb/Lilrb4a/Adora2a/icosl/Il12b/Ccl2/Ccl5/Rara/Ccr7/Cd300a/Cd83/Cyrb/Rac2/Socs1/Cd47/Cblb/H2-Ea/Sos1/Cd74/Malt1/Cd44/Il1b/Sirpa/Tox/Sit1/Zc3h12a/Runx3/Il4i1/Trex1
PRR signaling	22	Ly96/Ptgs2os/Esr1/Tnfaip3/Cactin/Tnfp1/Igtp/Cd300a/Nfkbia/Rftn1/Unc93b1/Slc15a3/Pik3ap1/Lyn/Smpd3b/Cd36/Tlr1/Tlr6/Oas3/Irak2/Cyba/Nlr1
Transcriptional factor activity	32	Sp100/Ptma/Tnfsf4/Aim2/Esr1/Tnfaip3/Zbtb7a/Cactin/Stat3/Psma6/Nfkbia/Edn1/Mtdh/Nfam1/Pim1/Tnf/Camk2a/Malt1/Tcf7l2/Traf1/Il1b/Hck/Crtc2/Zc3h12a/Cd36/Tlr6/Kit/Irak2/Bcl3/Slco3a1/Mapk3/Cx3cr1
Antigen presentation	15	Slc11a1/Mr1/Cd68/Ccr7/Psme2/Ext1/H2-Ea/Bag6/H2-D1/Rftn1/Cd74/Unc93b1/Clec4a1/Relb/Trex1

10- In Fig.6, only peritoneal cavity is tested as a non-lymphoid organ, I would be careful in generalising.

We have removed all wording about non-lymphoid organ in the text and figure 6 title (highlighted).

Quantifying the expression of EHF within DC subsets in the colon or tumor in disease settings as in Fig.1 would help with this. Similarly, quantifying the frequency/activation potential of mRegDCs vs other DCs in the same disease models in mice with DC-specific depletion of EHF would strengthen the role of EHF in mRegDCs.

We have now quantified the frequency and surface markers of CCR7⁺ DCs vs CCR7⁻ cDCs in DSS-colitis model. We found that EHF only regulates the phenotype of CCR7⁺ mregDCs but not CCR7⁻ cDCs (new **Supplementary Fig. 4d**).

11- EHF is expressed in different CCR7⁺ clusters (cluster 2 vs cluster 3) in the spleen vs LN (Fig6D). Do the authors know why this is the case?

Yes, there are some differences between cluster 2 and cluster 3 across organs. That is why we defined both cluster 2 and 3 as CCR7^{hi} DCs. Our hypothesis is that lymph nodes uniquely contain both resident and migratory DCs (higher CCR7), which differ in CCR7 expression and function.

Does it correlate with differences in CCR7/CD200/CD274 expression?

Yes, EHF expression is correlated to CCR7/CD200/CD274 (**Fig. 6c**, see below).

The authors present the overall integrated data in Sup figS6, but given the difference in EHF expression in LN vs spleen, it is important to look at those other markers per site too. This dataset could also be helpful in confirming other points.

Yes, the organs are different. We provided a more detailed analysis of all six samples (3 organs) in **Supplementary Fig. 6c**.

For example, do the authors see a correlation between TLR or IFN pathways and EHF expression within specific DC subsets?

We have added this data in the new **Supplementary Fig. 6d** (see below). The correlation did not seem to be too strong.

11- In Fig.6, stats on violinplots are done based on n being a cell. Since the authors have 6 samples, it should rather be done on a sample basis, to be sure that the phenotype is not driven by one sample.

Yes, the data for all six samples (3 organs) were presented in **Fig. 6d** (see below).

12- It should be noted that, while the authors focus on the potential inhibitory role of mReg DCs, whether they have a positive or negative effect on disease outcomes remains unclear, and enrichment of a CCR7⁺ DC signature is associated with improved survival in multiple cancers (Lee et al, Nat Comms, 2024). It would be interesting for the authors to discuss how their data integrate with this.

Thank you for sharing this information. First, this discrepancy might be the different clusters of CCR7⁺ DC. EHF is not expressed in CCR7^{lo} (Cluster 1), which is still considered to be CCR7⁺. This CCR7⁺ population in cancers might contain more CCR7^{lo} (Cluster 1). Second, EHF expression is downregulated quickly by GM-CSF and IFN γ in CCR7⁺ DCs after maturation. CCR7⁺ population in the more survivable “hot” cancers may not have sustained expression of EHF due to abundance of such cytokines. We added this in our discussion.

13- In some places, the number of independent experiments performed is not stated.

We have double-checked and made sure to correct any omission. Thank you!

Reviewer #3 (Remarks to the Author):

The study by Liu et al. identifies ETS homologous factor (EHF) as a key transcriptional regulator controlling maturation and immunosuppressive functions of conventional dendritic cells (cDCs) following TLR7 and TLR9 activation. Using conditional knockout mice, the authors demonstrate that EHF deficiency in DCs enhances resistance against DSS colitis, bacterial infections, and melanoma growth, primarily by promoting stronger Th1- and Th17-biased CD4+ T-cell responses. Mechanistically, the authors claim that EHF directly regulates the transcription of genes involved in dendritic cell maturation and immunosuppression, including *Ccr7*, *Cd200*, *Cd274* (PD-L1), *Irf4*, and *Rel*, thus highlighting its pivotal role in suppressing inflammatory responses.

This study represents the first comprehensive characterization of EHF's *in vivo* role in DC biology, expanding our understanding beyond its previously described functions in epithelial and tumor cell biology. The authors performed an extensive amount of experimental work, including the generation and analysis of new conditional knockout mouse models across several *in vivo* disease settings. Additionally, single-cell RNA sequencing analyses further reinforced their findings, providing high-resolution insights into EHF expression patterns across DC subsets in mice and humans at steady state and stimulatory conditions. A notable technical advance is the development of a novel monoclonal antibody recognizing EHF, which enabled CUT&TAG analyses to precisely identify putative EHF DNA binding sites.

While this manuscript provides novel insights into the role of EHF in DC biology and is supported by a big and diverse set of experiments, certain aspects of the study feel inconsistent, with some datasets conflicting or not fully aligning with the authors' claims. Some conclusions are stated too strongly relative to the presented data and would benefit from either additional experimental validation or a more nuanced interpretation.

Major points:

Figure 1:

My main concern is the limited analysis of DC2 subsets (cDC2a, cDC2b and DC3): although the data indicate that resident cDC2s express EHF at steady state, this EHF-expressing population was not examined or characterized in detail. Furthermore, DC2 subsets were not analyzed in sufficient depth in the *in vivo* models, either by flow cytometry or single-cell RNA sequencing. A more detailed investigation of DC2 heterogeneity in WT versus KO mice, both at steady state and in disease models, would greatly strengthen the manuscript's conclusions.

We have now conducted a more detailed analysis of DC subsets in steady state, CpG-B activation and colitis model (new **Fig. 4d-e**, see below). EHF regulated CCR7, CD200 and PD-L1 expression in cDC1s, cDC2As, and cDC2Bs.

Figure 1L: The authors challenge Ehf CD11c-Cre mice with B16-OVA and show that Ehf-deficient mice have increased survival compared to WT controls. They also demonstrate that, upon OT-I transfer, OT-I cells exhibit decreased PD-1 expression in Ehf-deficient mice, but they do not provide an explanation for this observation. Normally, PD-1 can act as both an activation and exhaustion marker, yet it remains unclear why OT-I cells show reduced PD-1 expression in the absence of EHF and what mechanisms drive this phenotype.

Thank you for pointing it out. We think that EHF-deficient DCs expressed lower levels of PD-L1 and that might contribute to the lower PD-1 expression on OT-I cells. We have now added this explanation (highlighted in text).

Furthermore, the authors should include data on the total numbers of engrafted OT-I cells

We have now included this data in the new **Supplementary Fig. 2d** (see below).

and perform tetramer staining to assess antigen-specific OT-I populations in WT versus Ehf-deficient mice.

We are sorry for the misunderstanding. OT-I cells are TCR transgenic that all share the same OVA-specific TCR, and they would be 100% tetramer positive. Thus, we do not think it is necessary to use tetramer staining on OT-I cells.

Figure S1J: How did the authors evaluate the expression of IFN α by pDCs using intracellular staining? The use of intracellular antibodies against IFN α is not described in the Materials and Methods section. This information is important, as intracellular staining for IFN α is technically challenging and has only been described by a few groups.

We have now added the details in the methods. We are proficient in IFN α staining since I was a postdoc with Dr. Boris Reizis, who is an expert on pDCs.

As an independent PI, I have published several papers which performed IFN α staining in pDCs:

Thioesterase PPT1 balances viral resistance and efficient T cell crosspriming in dendritic cells. **J Exp Med.** 2019. Sep 2;216(9):2091-2112

The RNase MCPIP3 promotes skin inflammation by orchestrating myeloid cytokine response. **Nat Commun.** 2021 Jul 2;12(1):4105

Cyclical palmitoylation regulates TLR9 signalling and systemic autoimmunity in mice. **Nat Commun.** 2024 Jan 2;15(1):1.

Figure S1O and S1Q: The authors need to include non-treated/non-infected mice as control groups.

We now added these control groups in the new **Supplementary Fig. 1o** and **S1q** (see below).

Figure 2:

Figure 2A-F: The study would benefit from an analysis that goes beyond cytokine production of OT-II engrafted cells by also assessing their true T helper cell identity through intracellular staining of GATA3, T-bet, Foxp3, and ROR γ t.

We have now confirmed the Th1 lineage via T-bet and Th2 lineage via GATA3 in **Fig. 2c-d** (see below). The results were consistent.

As mentioned in our methods, we utilized Foxp3⁺ staining as a standard definition of Tregs.

Furthermore, non-infected/non-treated mice should also be included as controls in this figure in the revised manuscript.

We have now added these controls in the new **Supplementary Fig. 2a** (see below).

Figure S2F-G: The authors show that Ehf-deficient mice have reduced inflammation when challenged with *C. rodentium*. This finding is unexpected and contrasts with the data shown in Figure 1/S1, where DSS-treated Ehf-deficient mice exhibited stronger inflammation compared to WT mice. Is the phenotype different if the authors analyze *C. rodentium*-infected mice at an earlier time point post-infection (e.g., day 6)? Could day 11 post-*C. rodentium* infection already be too late to observe the increased pro-inflammatory responses that might be expected in Ehf-deficient mice?

Thank you for pointing out this difference. We think that the DSS model mimics colitis, which is an autoimmune disease. In comparison, *C. Rodentium* is a pathogenic infection. Since EHF promotes both maturation and immunosuppression, it should have different outcomes in autoimmunity and infections.

In colitis, EHF-deficiency would result in less immunosuppression and mice would die quickly due to too much autoinflammation. In *C. Rodentium* infection, EHF-deficiency would result in stronger immunity and clear the pathogen faster, resulting in healthier mice with less inflammation.

Figure S2E: I would recommend moving Figure S2E into Figure S1, as it directly relates to the DSS experiment described in Figure 1 and would be better placed there for clarity and consistency.

We now moved it to Figure S1.

Figure 3:

Figure 3A: The authors performed qPCR on cDC1 and cDC2 stimulated in vitro with distinct TLR agonists to measure Ehf expression. They present the data as relative expression compared to the control sample. However, a more appropriate approach would be to normalize Ehf expression to a housekeeping gene. This would not only allow for comparison between the different stimulations but also provide an indication of the Ehf expression levels between cDC1 and cDC2.

This must be a misunderstanding based on the words we used. We do normalize it to a housekeeping gene first. In our original methods regarding qPCR, we stated: "All the data were normalized to β -actin expression." We have added the wording regarding housekeeping gene in Figure 3 legend accordingly. We are sorry for the confusion.

Figure 3B: The authors performed scRNA sequencing on splenocytes derived from B6 mice that were either treated with CpG-B or left untreated. In the CpG-B-treated panel, the cDC1 population does not appear to be annotated. Is this because cDC1 cells were absent in CpG-B-treated mice, or is this an annotation error?

Thank you for noticing this important point. After CpG-B treatment, scRNA-seq cannot differentiate CCR7⁺ DC1 and DC2, as all mature cDCs become just one cluster of CCR7⁺ DCs. Cluster analysis suggested that transcriptional programming of CCR7⁺ DC1 and DC2 are very similar. In FACS, it is still possible to differentiate them using traditional surface markers. However, there is a marked decrease in XCR1 expression by cDC1s (see below, also in Supplementary information, Gating strategy).

The sc-RNA-seq and FACS profiling suggests that the blurring of DC1/DC2 programming after CpG-B activation and EHF upregulation. This population of XCR1^{int} DCs may be of similar origin to the previously reported regulatory CD103^{int} DCs found in neonatal mice (*Sci. Immunol.* 2023).

This is also consistent with our finding that EHF repressed DC-lineage transcriptional factor such as IRF4, thus blurring the boundary between DC1 and DC2 when EHF is expressed (new **Fig. 4f**, see below). We have now added related discussion in the text (highlighted).

Figure 4:

Figure 4A: The authors demonstrate that cDCs derived from Ehf-deficient mice are reduced upon stimulation with CpG-B. It is important to elucidate which specific cDC subset is affected by the loss of EHF in this context. In particular, the authors should assess whether there is a decrease in cDC1, cDC2a, cDC2b, and/or DC3 populations.

We have now conducted a more detailed analysis of DC subsets in steady state and CpG-B activation (**Fig. 4c**, see below). There is a general decrease in cDC1s, cDC2As, and cDC2Bs.

Figure 4B-D: The authors demonstrate that cDC1 and cDC2 derived from Ehf-deficient mice show reduced expression of activation markers such as CCR7, CD200, and PD-L1, along with an increase in the transcription factor IRF4 and a decrease in c-Rel upon CpG-B stimulation. While these results are intriguing, they appear somewhat ambiguous, and it is unclear how the authors arrived at these findings. Performing bulk RNA sequencing on cDC1 and cDC2 from WT and Ehf-deficient mice treated with either CpG-B or PBS would help clarify these results and better capture the transcriptional changes associated with the absence of EHF.

Sorry, this must be a misunderstanding due to the order of the figures. In **Figure 6**, we conducted scRNA-seq analysis, which gave better clarity than bulk RNA-seq.

Figure 5:

Figure 5A-F: The authors performed a Cut&TAG experiments targeting EHF on BM DC cultures and used a broad population of cDCs, that contains both, cDC1 and cDC2 subsets to target EHF. Eventhough it gives already a good preliminary idea about the DNA binding regions of EHF; it still overlaps the peaks present in cDC1 and cDC2. A clearer and better approach would be to target ex vivo isolated cDC1 and cDC2 from the spleen that were stimulated with CpG-B or PBS. This would reveal the EHF binding sites in both, steady state as well as activation state and give a more and proufnd idea about how EHF regulates the transcriptional landscape within each cDC subset.

We have performed the CUT&TAG experiments on ex vivo isolated cDC1s and cDC2s from PBS or CpG-B treated mice as requested (**Supplementary Fig. 5f**, see below). The EHF bindings sites were similar in cDC1s and cDC2s.

However, due to the fragile state of ex vivo FACS-sorted cells, the background noise

was much greater than that of cDC^{FL-Notch}. Compared to traditional CHIP-seq, CUT&TAG has a more stringent requirement for cell viability. That was why we originally utilized cDC^{FL-Notch} to obtain a higher cell viability for lower background noise.

Figure 5E: The authors should tone down their statement (“...we found that EHF may directly promote the transcription of inhibitory molecules such as CCR7, CD200, and PD-L1.”, Line 491–492) regarding EHF specifically regulating the expression of these genes. Most of the depicted EHF DNA-binding sites are located within introns or in upstream/downstream enhancer regions, with only Cd274 showing a peak ~3 kb downstream of the promoter region. Since none of these enhancer regions were functionally evaluated, it cannot be conclusively claimed that EHF directly regulates these target genes.

We had performed motif mutagenesis in luciferase reporter assays on all putative ETS-binding sites. The results were confirmatory. These data were added in the new **Fig. 5g** (shown below). We hope this functional assay support our claim of direct transcriptional control.

(END)

Reviewer #1 (Remarks to the Author):

The authors have adequately addressed all reviewer concerns and questions, and the revision is acceptable.

As the mediator for Reviewer #3

Overall, the authors' responses to the reviewers' comments are satisfactory, and the major scientific concerns have been adequately addressed. I only note two minor issues where statements in the rebuttal do not appear to be reflected in the revised manuscript text, as detailed below.

1.The authors state in their response that an explanation linking reduced PD-L1 expression on EHF-deficient DCs to lower PD-1 expression on OT-I cells has been added to the manuscript. However, I was unable to identify an explicit statement to this effect in the revised text. Please indicate where this explanation is described or revise the text to clearly state this interpretation.

This statement is highlighted at line 260-262.

2.The authors state in their response that details of IFN- α staining have been added to the Methods; however, I could not locate any description of IFN- α staining in the revised manuscript or Supplementary Information. Please clarify where this information is provided, or revise the response accordingly.

This information is highlighted at line 560-570.

Reviewer #2 (Remarks to the Author):

The authors have responded to prior critiques. I do not have any further comments.